# Deep mutational scanning reveals the molecular determinants of RNA polymerase-mediated adaptation and tradeoffs

Alaksh Choudhury [1,2] ✉, Benoit Gachet[1], Zoya Dixit[1,3], Roland Faure [1,4,5], Ryan T. Gill[6,7] & Olivier Tenaillon [1,3] ✉

RNA polymerase (RNAP) is emblematic of complex biological systems that control multiple traits involving trade-offs such as growth versus maintenance. Laboratory evolution has revealed that mutations in RNAP subunits, including RpoB, are frequently selected. However, we lack a systems view of how mutations alter the RNAP molecular functions to promote adaptation. We, therefore, measured the fitness of thousands of mutations within a region of *rpoB* under multiple conditions and genetic backgrounds, to find that adaptive mutations cluster in two modules. Mutations in one module favor growth over maintenance through a partial loss of an interaction associated with faster elongation. Mutations in the other favor maintenance over growth through a destabilized RNAP-DNA complex. The two molecular handles capture the versatile RNAP-mediated adaptations. Combining both interaction losses simultaneously improved maintenance and growth, challenging the idea that growth-maintenance tradeoff resorts only from limited resources, and revealing how compensatory evolution operates within RNAP.

Protein evolution has been traditionally understood as the process of optimizing functions linked to a single trait[1]. However, over billions of years, proteins and protein complexes have also evolved to control multiple cellular traits. The molecular drivers for the evolution of multi-trait proteins are more complex than those of single-trait proteins, as multiple traits cannot be optimized simultaneously[2]. Improving a trait often leads to tradeoffs in others[3,4]. However, it is not known if tradeoff-associated traits are controlled by common residues in the protein, or if they are partitioned into independent sectors or modules[5]. An important example of a multi-trait protein complex is the RNA polymerase (RNAP), which not only performs transcription but also regulates cellular resource allocation through gene expression. In response to changes in the environment, the RNAP helps allocate resources for expressing genes pertinent to the environment and

determines the cell's phenotype[6]. Through regulation, the RNAP controls multiple traits that are associated with tradeoffs, as allocating resources to certain functions may deviate them from others[7,8]. For instance, under stringent regulation, the RNAP universally controls resource allocation between growth and maintenance[7,9].

The RNAP is the most frequently targeted protein complex for the adaptive laboratory evolution of *Escherichia coli* in diverse environments, such as growth in minimal media, high temperature, tolerance to industrial chemicals and antibiotics, long-term starvation, and silver nanoparticles; and even in cells with extreme modification such as genome size reduction by a million base pairs[10–19]. Therefore, the RNAP has garnered significant fundamental interest in understanding complex clinical phenotypes, and industrial interest for strain engineering in biotechnology and synthetic biology applications. RNAP is also a

[1]Université de Paris Cité, INSERM, IAME, UMR 1137, 75018 Paris, France. [2]Laboratoire Biophysique et Évolution (LBE), UMR Chimie Biologie Innovation 8231, ESPCI Paris, Université PSL, CNRS, 75005 Paris, France. [3]Université de Paris Cité, INSERM, CNRS, Institut Cochin, UMR 1016, 75014 Paris, France. [4]Université de Rennes, INRIA RBA, CNRS UMR 6074, Rennes, France. [5]Service Evolution Biologique et Ecologie, Université libre de Bruxelles (ULB), 1050 Brussels, Belgium. [6]Renewable and Sustainable Energy Institute (RASEI), University of Colorado-Boulder, Boulder, CO 80309-0027, USA. [7]Novo Nordisk Foundation, Denmark Technical University, 2800 Kgs Lyngby, Denmark. ✉e-mail: alaksh.choudhury@espci.fr; olivier.tenaillon@inserm.fr

target for essential antibiotics, such as Rifampicin against tuberculosis. The fitness costs of the rifampicin resistance-conferring clinical RNAP mutations may promote the emergence of extreme drug resistance and multidrug resistance strains[20–22]. Therefore, understanding RNAP-mediated evolution also has broad industrial and clinical significance.

Despite its ability to allow rapid adaptation in multiple environments, the sequence of the RNAP is highly conserved across all domains of life[23]. Moreover, during ALE, the selected mutations specifically target residues that are broadly conserved[24]. According to a dominant hypothesis, RNAP mutations rewire global transcription to match the expression requirements for the specific selection environment[25]. However, RNAP mutations are also often highly pleiotropic i.e., selected RNAP mutations have benefits across multiple conditions. While some mutations in RNAP can improve growth on multiple sugars[3], others can promote cross-stress resistance[12,26]. Alternatively, the mutations are also often involved in trade-offs[27]. While improved growth can increase pH sensitivity, decrease motility, and antibiotic persistence[3]; RNAP mutation-mediated improved stress tolerance can decrease growth[12,14]. Due to the pleiotropy, it is difficult to identify the specific cellular traits under selection and delineate the true systems' impact of adaptive RNAP mutations.

Previous studies have characterized a few RNAP mutations at systems and molecular scales. At a systems level, RNAP mutation-mediated adaptation and trade-off may be associated with a global rebalancing of resources[3]. For example, transcriptomic analysis of a couple of RNAP mutations, which improved growth in minimal media with different sugars, revealed a global rewiring of transcription to favor growth-associated functions at the cost of genes associated with stress response[3]. The global rewiring of transcription could be linked to molecular changes in the RNAP which favors the expression of one class of genes over the others. For instance, a few mutations selected for improved stress tolerance and long-term starvation fall into a category of RNAP mutations called "stringent" mutations[12,14]. Some RNAP stringent mutations are known to impact the stability of one of the open complex intermediates to favor the expression of stress-associated genes over growth[28]. Would that mean, the growth-improving mutations also act on the open complex stability? Although the RNAP has been extensively characterized structurally and biochemically in vitro[29], its subfunctions, and interactions are not clearly associated with systems traits. Studying multiple mutations in multiple environments can increase the predictive power of their systems' effect and identify tradeoffs[27]. However, due to a paucity of the characterized mutations, it is not known if tradeoff-associated traits are controlled by the same or different molecular subfunctions; and if and how systems-level tradeoffs have influenced the evolution of the RNAP.

To build an RNAP molecular function–systems phenotype map, we can characterize a large number of RNAP variants using high-throughput deep mutational scanning or DMS. In DMS, the fitness of thousands of variants is measured concurrently for a sequence of interest[30]. Upon subjecting the library of variants to a selection pressure, the frequency of variants changes due to their associated fitness. The frequencies of beneficial variants increase and the deleterious ones decrease. A log change in the variant frequency (measured using deep sequencing) relative to the wildtype sequence provides a fitness estimate. Mapping the fitness scores on the protein sequence and structure provides significant insights into protein function and evolution (Fig. 1a). When protein fitness is associated with cellular reproductive success, DMS also provides insights into the systems impact of protein mutations; such as collateral growth effects of protein misfolding, cost of resistance, and effects of metabolite flux and toxicity[31–33].

It is important to perform DMS for essential protein complexes, such as the RNAP, in their native genomic context. Plasmid-mediated DMS can be confounding due to changes in gene expression, copy number effects, and loss of epigenetic regulation[34]. We recently developed CRISPR/Cas9-mediated genomic error-prone editing

(CREPE) technology for deep mutational scanning of essential genes, in their native genomic context in *E. coli*[32]. CREPE exploits Cas9-mediated recombineering to replace a genomic target with an error-prone PCR library to develop a variant library with high mutation efficiency (~50–80%)[32]. The rich diversity allowed precise fitness estimates to gain a mechanistic and biochemical understanding of antibiotic resistance targets[32].

Here we apply CREPE to understand molecular determinants of RNAP-mediated adaptation and tradeoffs (Fig. 1a). By analyzing adaptive mutation databases, we identified a region within the RpoB subunit of the RNAP where mutations improved fitness in multiple environments. We used CREPE to generate a rich library of 6000 variants with mutations targeting this region and measured the fitness of the variants in multiple environments. According to the dominant GTME hypothesis, mutations in the RNAP allow for condition-specific adaptation. We mapped the fitness, measured at a residue-level precision, to the target sequence and structure, and identified residue clusters important for adaptation in different environments (Fig. 1a). Adaption in some environments was associated with tradeoffs in others. So, we were able to evaluate if tradeoff-associated traits were controlled by common or modular clusters of residues. Consequently, we gained a mechanistic understanding of RNAP-mediated adaptation and trade-offs; and crucial insights into the molecular drivers of complex protein evolution.

## Results
### A hotspot for adaptive mutations occurs in the RNAP
Recently two databases, the Resistome, and the *E. coli* mutation database cataloged 5000 and 15000 mutations identified in adaptive laboratory evolution respectively[17,18]. According to both studies, the RNAP and more specifically the β subunit of the RNAP (RpoB) is the most frequently targeted site for adaptive laboratory evolution to diverse stresses[17,18]. Mutations in *rpoB* were selected in ~31% of the 178 evolution conditions; the most diverse compared to any other gene[17]. We found that 40% of all known RNAP mutations were concentrated within a 100 amino acid region between the positions 500–600 (Fig. 1b). Mutations within this "target" region were selected in more diverse environments compared to any other region of the RNAP. The conditions include improved fitness in minimal media with multiple sugars such as glucose[3], lactate[10], and glycerol[3], and in the presence of different stresses including high osmolarity, hydrogen peroxide, and butanol[26], high temperature[11], antibiotic resistance and cross-resistance[19], and silver nanoparticles[15] (Fig. 1b). Due to its ability to improve fitness in diverse environments, the target region is an ideal model to study RNAP-mediated adaptation.

The region occurs close to the catalytic core of the RNAP and contains residues that interact with the DNA (non-template strand) and the RpoC subunit of the RNAP (Fig. 1c). We estimated the conservation score for RpoB using the consurf algorithm[35]. A higher Consurf grade corresponds to high conservation and functional importance. The target region had the highest Consurf score compared to the entire RpoB protein (Krushkal Wallis H-test, KWH-test, *p*-value < 10^−16) (Fig. 1d). Therefore, the target occurred in a highly conserved and potentially functionally important region of the RpoB.

Using CREPE, we developed a library of ~6000 variants in the target in wild-type *E. coli*, with single and multiple mutations per variant (Supplementary Fig. 1a, b). We estimated the variant fitness in five conditions with different sugars and stresses: M9 minimal media with glucose, glycerol, and galactose, M9 minimal media with glucose and high osmolarity (0.3 M NaCl), and M9 minimal media with glucose and 0.6% Butanol (Supplementary Fig. 1c, d, Supplementary Table 1). We chose conditions in which RNAP mutations had been previously reported. During the evolution, the variant frequency increases for beneficial mutations and decreases for deleterious ones (Fig. 1a). We measured the variant fitness as the slope for the log change in variant

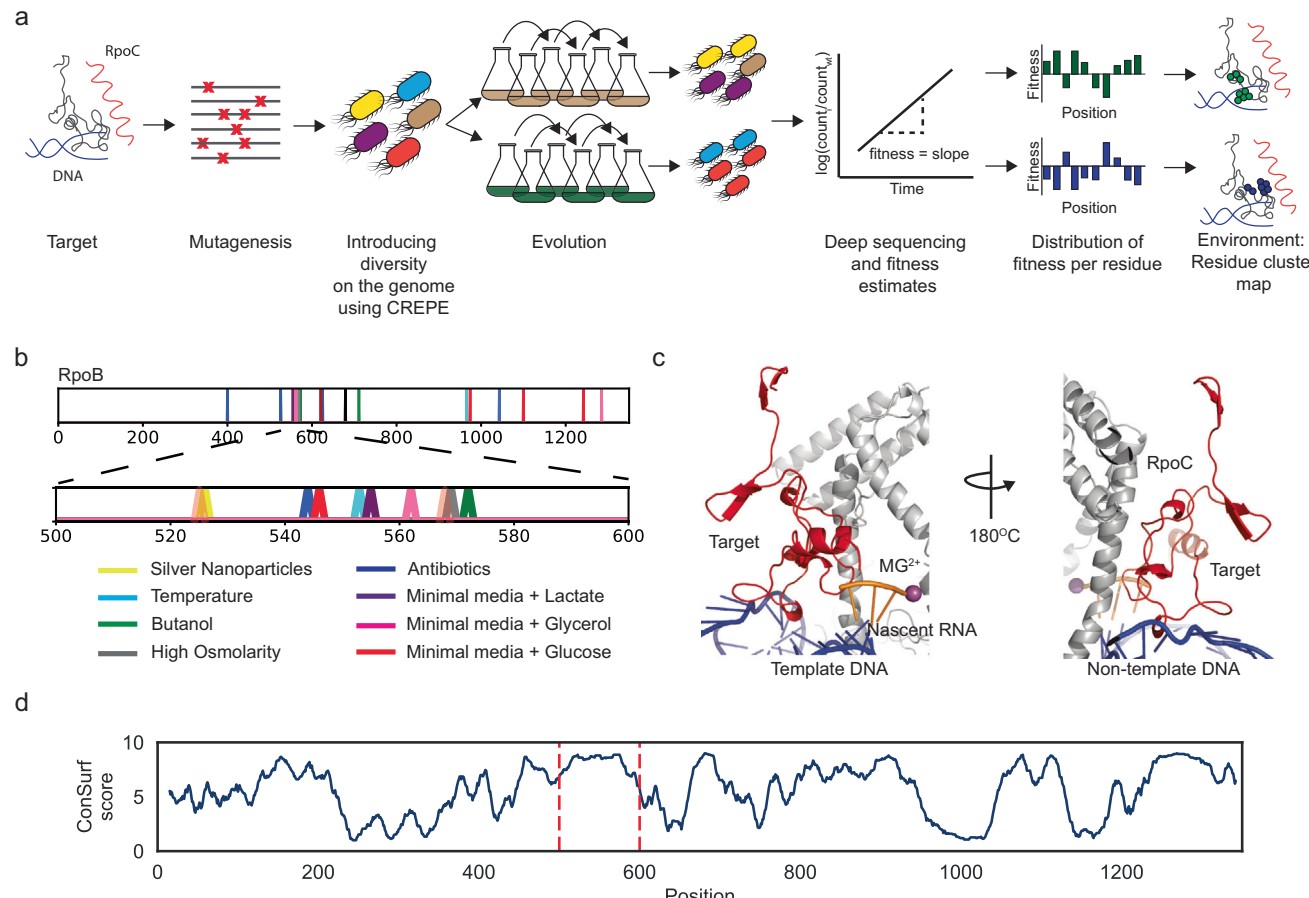

**Fig. 1 | Mutations in the "target" sequence of the RNAP β subunit (RpoB) improve fitness in diverse conditions. a** CREPE-based experimental setup to estimate the fitness of RNAP mutations in multiple environments. **b** An adaptive evolution mutation map of RpoB depicting mutations identified in different environments (colored peaks), with the target region highlighted. Conditions corresponding to the peaks have been highlighted below the figure. **c** The target region (red cartoon) in the RNAP is proximal to the catalytic core (pink $MG^{2+}$ sphere), DNA (blue), and the RpoC subunit (the Bridge-Helix) (Gray). **d** The rolling average Consurf score (blue line, averaged over a 15 amino acid window size) for RpoB. The target region occurs between the dashed red lines. Source data are provided as a Source Data file.

frequency, relative to a wildtype control (material and methods). We validated the fitness using multiple methods (Supplementary Note 1). To make inferences using mean growth-associated fitness and mean stringent enrichment, we also verified that we sampled substitutions with a broad range of physio-chemical properties in the target (Supplementary Note 1 and Supplementary Fig. 1e, f).

**Beneficial mutations show cross-environment adaptation**

RNAP mutations are often pleiotropic as improved fitness in one environment can impact fitness in other environments. At position 526, the same mutation H526Y improves fitness in the presence of antibiotics and silver nanoparticles[15,19] (Fig. 1a). At positions 545 and 546, the same mutation improves growth in minimal media with different sugars such as glucose, glycerol, and xylose[3]. Similarly, an osmotolerant mutation at position 569 also improves fitness in the presence of butanol, hydrogen peroxide, and at low pH[26]. RNAP mutations also have trade-offs. Growth-improving mutations at residues 545 and 546 show decreased motility, decreased antibiotic persistence, and decreased pH resistance[3]. We evaluated the extent of cross-environment adaptation or tradeoffs by correlating the fitness of the variants between environments. A correlation of fitness for beneficial mutations between environments would suggest cross-environment adaptation. Alternatively, positive fitness in one environment and negative in the other, to make a Y-shaped correlation, would signify the presence of a tradeoff (Fig. 2a). The variant fitness was strongly

positively correlated across environments (Fig. 2b). Therefore, the beneficial target mutations were generalist because they improved fitness in multiple environments. The beneficial mutations likely impacted a common global trait, as opposed to condition-specific traits. We individually reconstructed several beneficial mutations and found that each beneficial mutant had an increased growth rate in the minimal media used for the adaptation experiment (Supplementary Fig. 2a, b). Therefore, we posited that the target region was involved in RNAP-mediated growth control.

**The target region is involved in growth control and stringent response**

The growth rate is regulated by many mechanisms including the stringent response[8]. In fast-growing cells, most of the cellular RNAP transcribes growth-associated genes such as ribosome biosynthesis genes. Upon experiencing stresses such as starvation, heat shock, and nutrient downshifts, the stringent response redirects the RNAP by inhibiting the transcription of growth-associated genes and activating the expression of maintenance genes such as amino acid biosynthesis[9]. We hypothesized that the target region may be involved in stringent regulation. The lag i.e., the time required to adapt to nutrient downshifts is determined by the stringent response[3,36]. We found that all reconstructed growth-improving RpoB mutants had significantly increased lag times for glucose to acetate transition (Supplementary Fig. 2c). Multiple growth-favoring RNAP mutants also had a delayed

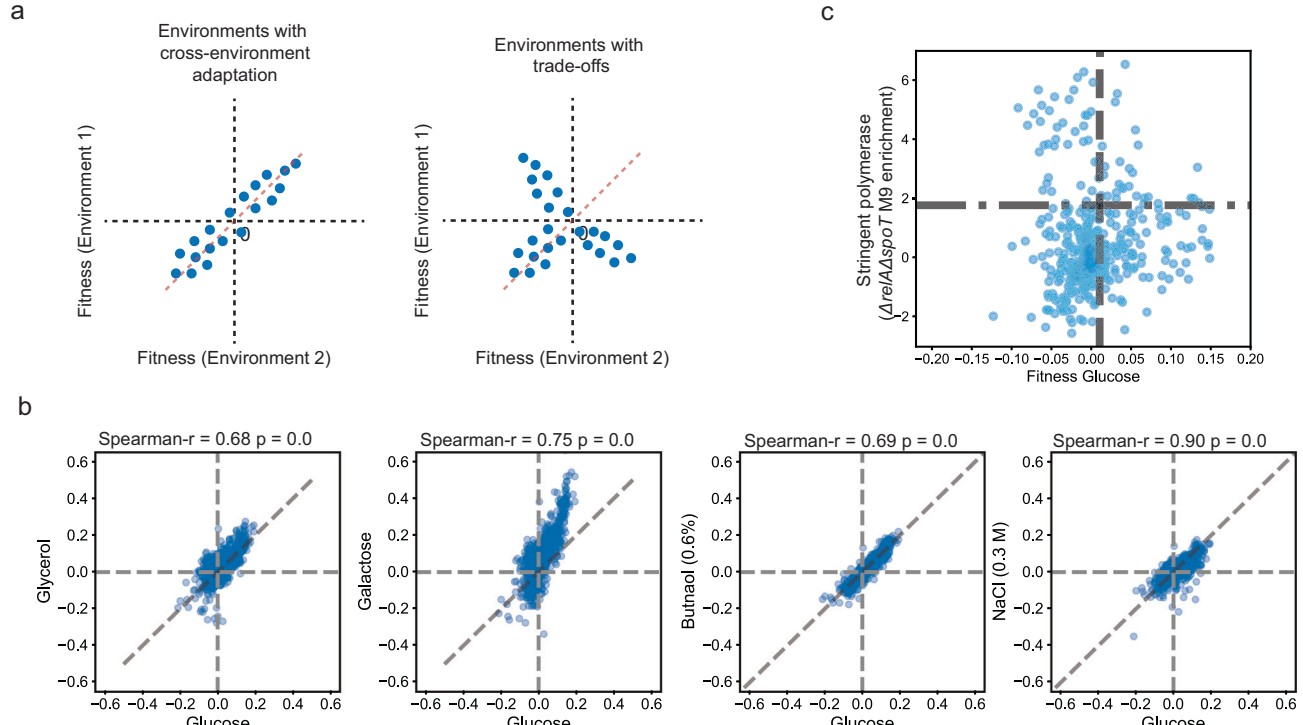

**Fig. 2 | Correlation of fitness between environments to identify cross-stress adaptation and tradeoffs for target mutations. a** Expected correlation of fitness of variants between environments in the case of cross-environment adaptation (left) or tradeoffs (right). **b** Correlation of the fitness for target mutations measured in M9 minimal media with glucose to the fitness measured in M9 media (blue spheres, left to right) with glycerol, galactose, glucose, and 0.3 M NaCl, glucose, and 0.6% (v/v) butanol respectively. The Spearman correlation coefficient and the associated *p*-value are mentioned at the top of each curve. **c** Correlation of fitness for different variants within the target region measured in M9 minimal media with glucose (blue spheres) with the enrichment of the same variants in the *ΔrelAΔspoT* strain of *Escherichia coli* upon plating on M9 minimal media agar with Glucose (see Supplementary Fig. 3 for detailed information). Source data are provided as a Source Data file.

induction of maintenance-associated (stringent response-induced) amino acid biosynthesis gene *hisG* compared to the wild-type RNAP (Supplementary Fig. 2d).

Therefore, we postulated that the target may also contain residues where mutations favor the stringent response and reduce growth. Stringent mutations of the RNAP can be selected by imposing amino acid starvation in the *ΔrelAΔspoT* strain of *E. coli*[28]. When *E. coli* experiences external stress or starvation, there is an increase in the concentration of the small molecule alarmone ppGpp. ppGpp and the transcription factor DksA bind to the RNAP to induce conformational changes in the RNAP[9]. These conformational changes activate the transcription of genes important for maintenance/starvation response such as amino acid biosynthesis[9]. Therefore, null ppGpp strains, such as *ΔrelAΔspoT* strain of *E. coli*, with both ppGpp synthesis enzymes (RelA and SpoT) deleted, are auxotroph for amino acids. *ΔrelAΔspoT* strains of *E. coli* cannot grow in minimal media in the absence of amino acids. Stringent mutations of the RNAP mimic the ppGpp-bound state to escape the auxotrophy by constitutively activating the amino acid biosynthesis genes (Supplementary Fig. 3a)[28]. Consequently, the stringent mutations of the RNAP favor maintenance/starvation-associated functions. We constructed a library of variants within the target region in the *ΔrelAΔspoT* strain of *E. coli*. We observed significantly more CFUs for the RpoB library in the *ΔrelAΔspoT* strain compared to a non-edited control upon plating on minimal media without amino acids (Supplementary Fig. 3b). We measured the variant stringent enrichment (fitness) as a log change in variant frequency before and after plating on M9-Glucose relative to the wild-type control (Supplementary Fig. 3c). We identified 123 single (and ~418 total) stringent mutations within the target, including previously described stringent mutations (Supplementary Fig. 3b, c and Supplementary Note 1, 2,

Supplementary Table 2). The presence of stringent mutations within the target suggested its role in the stringent response.

The stringent response regulates resources between growth and maintenance. It is proposed that the resources are "limited". Consequently, favoring one objective is proposed to be associated with a tradeoff with the other[7] (also seen above for some growth-improving mutations, Supplementary Fig. 2c, d). Upon correlating the growth-associated fitness and stringent enrichment, we observed a tradeoff-associated Y-shaped correlation (Fig. 2c). However, contrary to expectation, we observed some mutations that were both stringent and had significantly improved growth as well (Fig. 2c).

## Modular residue clusters determine the growth and stringent phenotypes

Some stringent mutations of the RNAP within the target are known to decrease the stability of an initiation open complex intermediate to inhibit growth-associated genes and activate amino acid biosynthesis[28] (Supplementary Table 3). Due to the observed tradeoffs, we hypothesized that both growth-improving and stringent mutations within the target could be associated with open complex stability-determining residues. We mapped the mean growth-associated fitness and mean stringent enrichment on the target sequence. As opposed to our hypothesis, the residues with high growth-associated mean fitness and the ones with high mean stringent enrichment rarely coincided (Fig. 3a, b). Rather we observed a residue-level specificity for multiple positions i.e., if mean growth-associated fitness was high, the mean stringent enrichment was low and vice versa (Fig. 3a).

To uncover the molecular mechanisms underlying these selections, we analyzed the organization of the fitness-conferring mutations

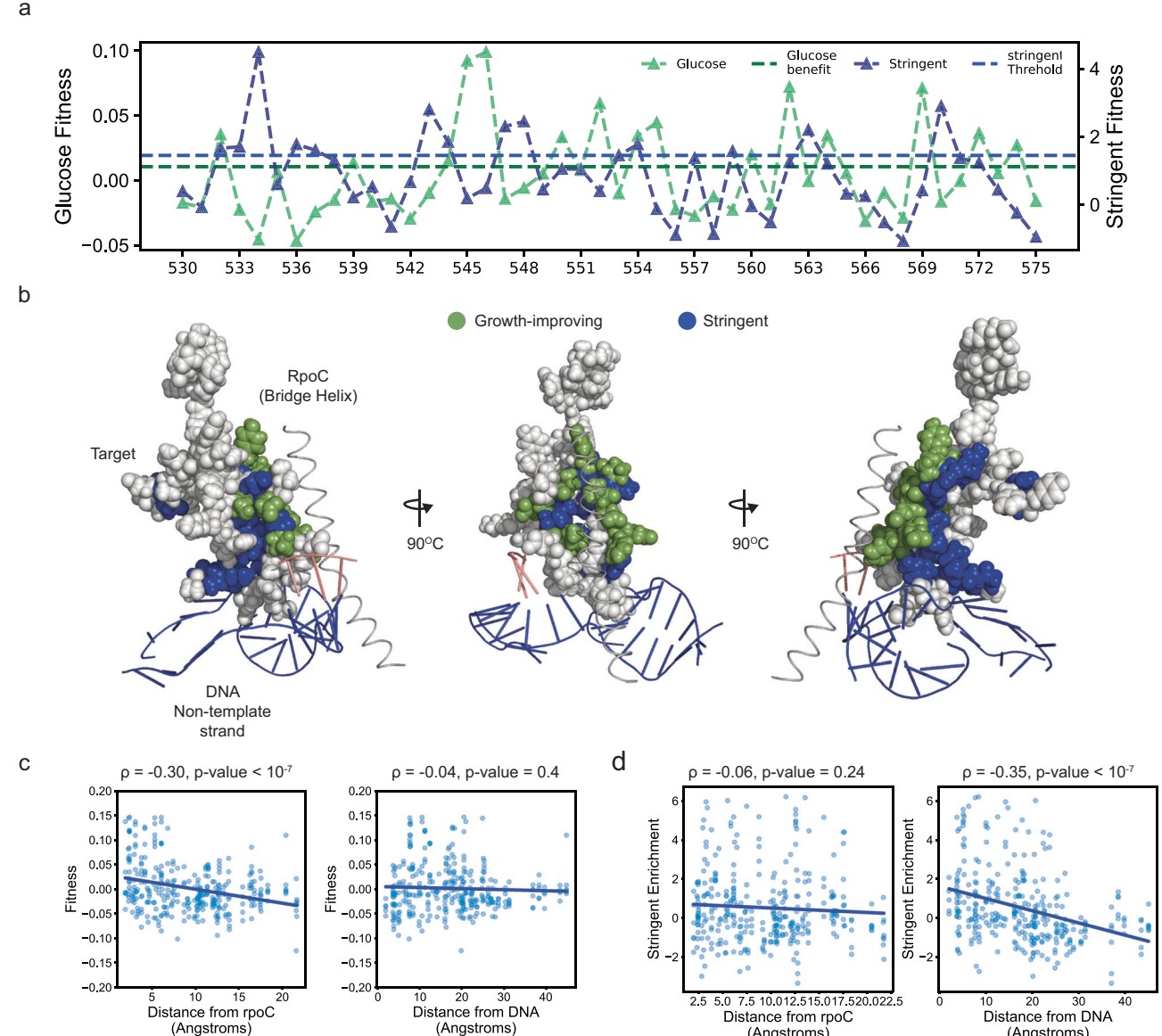

**Fig. 3 | Modular residues clusters are important for growth and stringent response. a** Residue-wise mean growth-associated fitness for all mutations (green triangles, y-axis on the left) and mean stringent enrichment (blue triangles, y-axis on the right). **b** Residues with high mean growth-associated fitness (green spheres) and high mean stringent enrichment (blue spheres) were mapped on the structure of the target region. **c** Correlation between growth-associated fitness and the minimum distance of the residue from the β′ RNA polymerase subunit (RpoC) (left), and growth-associated fitness and the minimum distance of the residue from the DNA (RpoC) (right). **d** Correlation between stringent enrichment and the minimum distance of the residue from the β′ RNA polymerase subunit (RpoC) (left), and growth-associated fitness and the minimum distance of the residue from the DNA (RpoC) (right). Source data are provided as a Source Data file.

within the RNAP structure. The target region interacted with the RNAP RpoC subunit and the DNA (Fig. 3b). Univariate and multivariate analysis revealed that the growth-associated fitness was negatively correlated with distance from RpoC but not with distance from DNA (Fig. 3c and Supplementary Table 4). In contrast, the stringent enrichment was negatively correlated with distance from DNA but not with distance from RpoC (Fig. 3d and Supplementary Table 4). Despite being associated with sequentially and structurally proximal residues, each phenotype may be controlled largely by a specific set of residues, likely associated with different molecular interactions.

**Internal target or target-DNA interactions determine the stringent phenotype**

We identified 123 unique single stringent mutations of the RNAP distributed across the target region (Fig. 4a). Stringent mutations of the RNAP alter the stability of an initiation open complex intermediate[28].

Amongst several other factors, the stability of the open complex is determined by interactions with the non-template strand of the DNA[37]. Accordingly, mutations in several DNA-proximal residues 532-534, 536-538, and 543-544 led to the stringent phenotype (Fig. 4b and Fig. 4c). In this cluster, substitutions in the hydrophobic residues to amino acids with polar, charged, and bulky side chains, which could interfere with DNA interactions, led to the stringent phenotype (with significantly higher stringent enrichment compared to both synonymous mutations and hydrophobic residues KWH-test, $p$-value $= 10^{-8}$ and $10^{-3}$ respectively, Supplementary Fig. 4a, b). The cluster contained a beta-turn with the Glycine 534. Unlike other hydrophobic residues, mutation of this glycine to any other residue types (hydrophobic, polar, or charged), resulted in stringent polymerases (Fig. 4c). The glycine residue is important for the folding in the beta-turn motif[38]. This suggested that residue 534, and likely the beta-turn motif, played a role in the open complex intermediate stability.

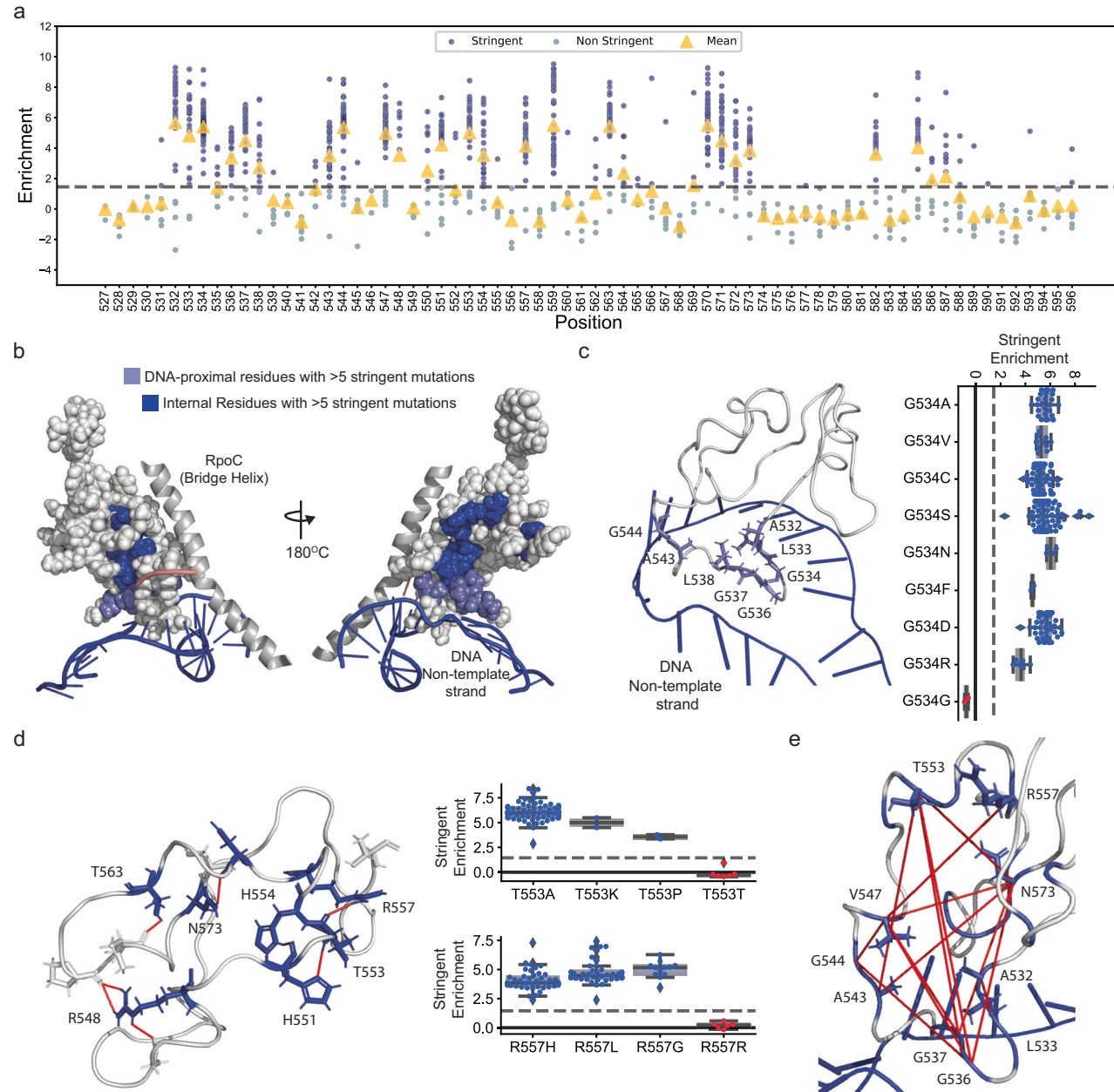

**Fig. 4 | Stringent mutations within the target. a** Residue-wise distribution of stringent enrichment for each variant (dots). Variants (blue dots) above the cut-off (grey dashed line, Supplementary Fig. 3C) are stringent mutations of the RNAP. The yellow triangles represent the mean enrichment score. **b** Internal (dark blue) and DNA-proximal (light blue) target residues with greater than five unique stringent mutations mapped on the target structure. **c** (Left) DNA-proximal residues with high mean stringent enrichment (blue sticks). (Right) A bar plot (blue) of the stringent enrichment score of non-synonymous (blue) and synonymous (red) mutations in the residue G534 relative to the cutoff stringent-enrichment score (grey dashed line, Supplementary Fig. 3). Within each box, the horizontal black lines represent median values, lower 25th percentile and upper 75th percentile bounds, and the whiskers represent extreme values in the 1.5× interquartile range. There were 28, 10, 64, 122, 4, 2, 40, 6, and 2 independent observations for variants top-bottom in the plot. An observation represents an independent fitness measurement of a synonymous variant of the focal mutation in two biological replicates

(Supplementary Note 1). **d** (Left) Residues with high mean stringent enrichment (blue sticks) with internal polar interactions (red lines). (Right) A bar plot (blue) of the stringent enrichment score of non-synonymous (blue) and synonymous (red) mutations for some highlighted residues relative to the cutoff stringent-enrichment score (grey dashed line, Supplementary Fig. 3). Within each box, the horizontal black lines represent median values, lower 25th percentile and upper 75th percentile bounds, and the whiskers represent extreme values in the 1.5× interquartile range. There were 70, 2, 2, and 6 independent observations for T533 variants and 44, 42, 12, and 5 independent observations for R557 variants left to right in the plot. An observation represents an independent fitness measurement of a synonymous variant of the focal mutation in two biological replicates (Supplementary Note 1). **e** Residues with high mean stringent enrichment (blue) with red lines connecting two residues with significant positive epistatic interaction (Supplementary Fig. 4 and Supplementary Note 2). Source data are provided as a Source Data file.

Mutations in another set of internal residues also led to the stringent phenotype (Fig. 4b). Within this cluster, a mutation in the tyrosine residue at position 563 is known to decrease open complex intermediate stability[28]. We observed that the side chain of the tyrosine

residue had a predicted internal polar interaction within the target region (Fig. 4d). Mutations that would disrupt the internal polar interaction at position 563, including the open complex intermediate-destabilizing mutation, were stringent (Supplementary Fig. 4c).

Similarly, in several other residues 548, 551, 553, 554, 557, 563, and 573 had predicted internally-interacting polar side chains and mutations that disrupted the polar interactions were stringent (Fig. 4d and Supplementary Fig. 4c). Therefore, the internal polar interactions also determined the stringent phenotype by controlling the open complex intermediate stability.

The internal residues, such as 563, that affected the open complex intermediate stability were far away from the DNA-interacting residues. We posited that the distant residues possibly interacted with the DNA-proximal residues. Previous work has also suggested allosteric interactions between RpoC (Bridge Helix)-proximal and DNA-proximal residues to effectuate stringent response[39]. Possible long-range interactions can be inferred by measuring epistasis in double mutants. We estimated the epistasis for all double mutants and evaluated if distant residues had significant positive epistasis between them (Supplementary Note 2, Supplementary Fig. 4d, e). We found significant positive epistasis between the internal and DNA-proximal residues, suggesting the presence of such long-range interactions (Fig. 4e, Supplementary Note 2, and Supplementary Fig. 4d, e).

### Target-Bridge Helix (BH) interaction determines the growth phenotype

To understand the growth phenotype, we first looked at the Distribution of Fitness Effects or the DFE. The DFE is a histogram of the frequency of the measured fitness effect size. A DFE almost universally shows an exponentially diminishing positive tail because beneficial mutations are rare and account for ~0.01–1% of mutations[1,40]. On the contrary, the DFE for the growth-associated fitness had a prominent beneficial mutation mode (Fig. 5a). Amongst all unique single mutations scored, 28.4% were beneficial, 39.7% were neutral, and 31.9% were deleterious. It is important to mention here that, since the RNA polymerase is an essential gene, we cannot score null mutations. However, even after excluding an average 20 to 30% null mutations by comparing different DFEs, the fraction of beneficial mutations should not exceed 2%. Therefore, the large number of beneficial mutations suggested a contrast between the protein-level and cellular-level impact of the mutation. We posited that a partial loss of protein-level function/interaction likely improved cellular-level fitness. Therefore, many mutations that would appear in the (usually large) deleterious/slightly deleterious mode of the DFE may have shifted to create the prominent beneficial mode. We observed that at several positions all or almost all (>95%) substitutions increased fitness (maroon arrows, Fig. 5b). Since, at any position, substitutions are more likely to be deleterious than beneficial, the observation further suggested that the improved fitness may be associated with a partial loss of a subfunction or interaction.

Several but not all residues with high mean growth-associated fitness clustered around the β-β' (RpoB - RpoC) interaction surface (Fig. 5c). The target region was proximal to a functionally important motif in the β' subunit of the RNAP, the Bridge Helix (BH). Target residues with high fitness mean clustered around the highly conserved, and known functionally-important residues of the BH[41] (Fig. 5d). Two conserved positively charged residues, H777 and R780, in the bridge helix were in proximity to residues with high mean growth-associated fitness (Fig. 5e). In each case, substitutions of the proximal residues to positively charged residues, which would decrease target-BH interaction due to repulsion to the positively charged BH residues, significantly increased growth-associated fitness (Fig. 5e). Alternatively, interaction-increasing substitutions, N573D and V558E, to negatively charged residues significantly decreased the growth-associated fitness (Fig. 5e). The residue F545 occurred in close proximity to a lysine, K789, residue. Aromatic rings close to positively charged residues can form cation-pi interactions. Mutations that may disrupt the possible interaction also improved growth-associated fitness (Supplementary Fig. 5a). Therefore, decreased target-BH interaction possibly improved growth.

We used the antibiotic CBR703 to further validate if target-BH interaction affected growth. The antibiotic CBR703 binds in the RpoB-BH interaction interface and may strengthen the interaction to affect RNAP function[42]. CBR703-resistant mutations decrease the RpoB-BH interaction strength[42]. Therefore, if decreased target-BH interaction increased fitness, resistance to CBR703 would correlate with growth-associated fitness. The MG1655 strain has a very high minimum inhibitory concentration (MIC) to CBR703[43]. Deleting the *tolC* gene increases sensitivity to CBR703[43]. We reconstructed the *rpoB* variant library in the *Keio ΔTolC* strain. We found 205 single CBR703-resistant mutations in the *Keio ΔtolC* strain (Supplementary Fig. 5b, c). CB5R703 resistance, estimated as the log fold- enrichment for growth in CBR703, correlated strongly with the growth-associated fitness (Fig. 5f, Keio: $\rho = 0.5$ and $p$-value $< 10^{-16}$); and all except one resistance-conferring residue had high mean growth-associated fitness (Fig. 5g).

Mutations in residues far away from the target-BH interaction interface such as 572, 574, 532, and 535 (20 Å from BH) increased growth and caused CBR703 resistance (Figs. 5c, g). Similar to the stringent phenotype, we observed significant positive epistasis between the buried residues and the BH-adjacent residues (Fig. 5h and Supplementary Fig. 5d, e). Therefore, long-range interactions likely altered the BH-target interaction-controlled phenotype to affect growth and CBR703 resistance.

Biophysically, CBR703-resistant RNAP variants, have increased catalysis rate and resistance to regulatory pauses[42]. The correlation between CBR703 resistance and growth suggested that improved growth was associated with increased catalysis and pause resistance. The BH controls catalysis by interacting with RpoB and the Trigger Loop in the RpoC subunit[44]. Mutations in both the BH and Trigger loop, have been selected for improved growth in ALE experiments (Supplementary Fig. 6a). To further test the hypothesis, we constructed two mutations in the RpoC subunit, I774T and I755V, known to increase catalysis rate and pause resistance[42] (Supplementary Fig. 6b). We observed that both mutations significantly improved growth compared to the wild-type variant. This further validated that improved growth may be associated with the increased transcription rate and pause resistance

### The growth and stringent phenotypes can be achieved simultaneously

Mutations at certain residues both improved growth and led to the stringent phenotype (Fig. 2c). Each phenotype is associated with residues involved in different interactions. The stringent phenotype was associated with a loss of internal polar interactions (Fig. 4) and the growth phenotype with decreased target-BH interaction (Fig. 5). We next asked if mutations leading to the combined phenotypes affected both interactions. The residue N573 had the highest number of such mutations with the combined stringent-growth phenotype. The residue 573 had a predicted internal polar interaction with residue C559 and was near a positively-charged BH residue R780 (Fig. 6a). Mutation of N573 to target-BH interaction-decreasing positively charged residues, N573K and N573H, improved glucose-associated fitness, and that to a negatively charged residue, N573D, decreased fitness (Fig. 6b). On the contrary, the three mutations N573K, N573H, and N573D, which would break the polar interaction were stringent. We reconstructed the N573K variant and confirmed that the variant had improved growth and was stringent (Supplementary Fig. 7). Therefore, the loss of both interactions leads to both improved growth and the stringent phenotype, suggesting that each objective could be optimized simultaneously.

To further verify the independent optimization of each trait, we next looked at the double mutations. We first looked at double mutations combining a stringent and a growth-improving mutation. A majority, 70.3% (64/91), of such combinations had both the growth-improving and the stringent phenotype (Blue, Fig. 6c).

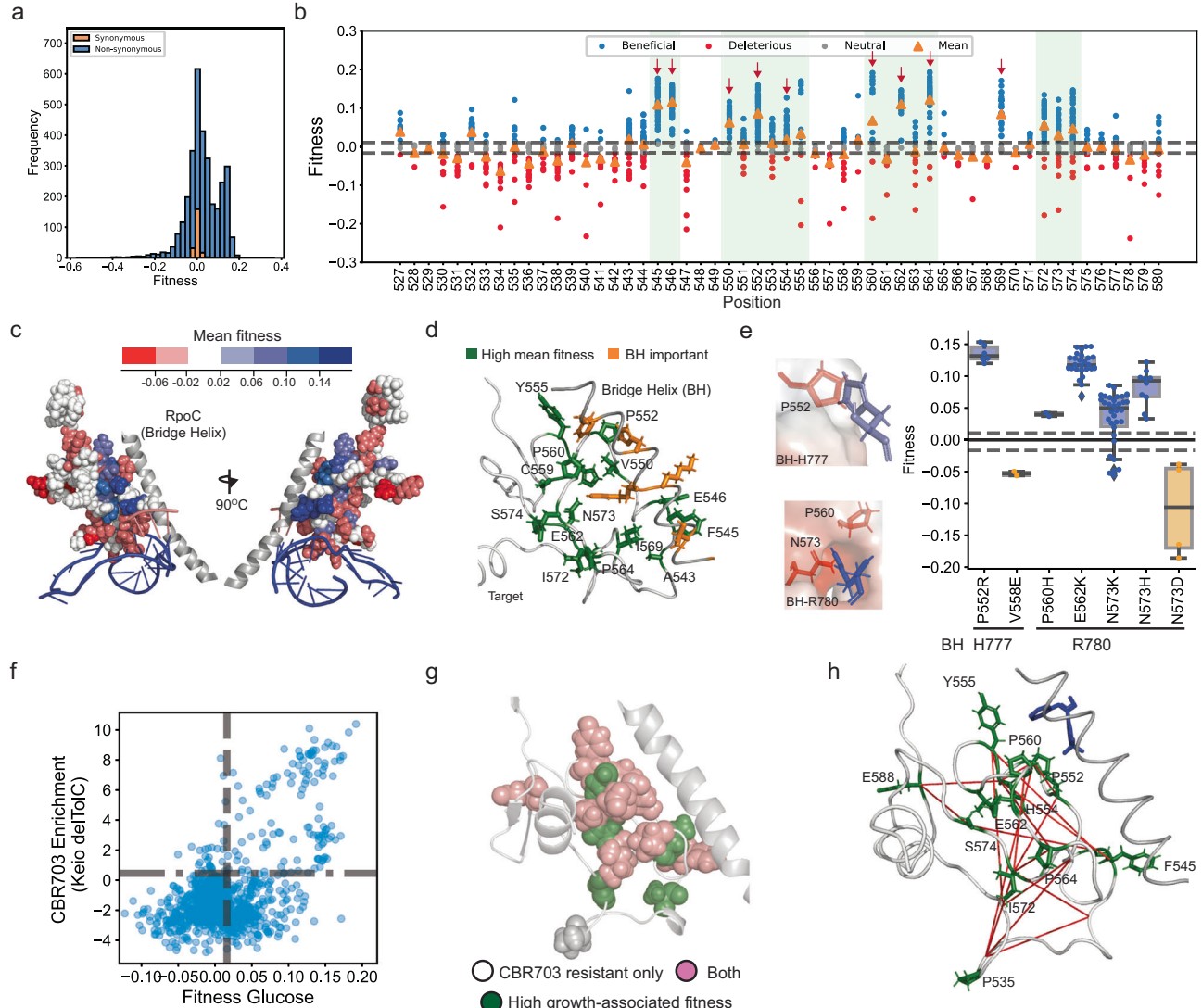

**Fig. 5 | Growth-improving mutations within the target. a** Distribution of fitness effects for the growth-associated fitness measures in M9 minimal media + Glucose for all (blue) and synonymous mutations (orange). **b** Residue-wise distribution of growth-associated fitness for beneficial variant (blue dots), synonymous variants (gray dots), and deleterious variants (red dots) with the cut-off for beneficial fitness score (top gray line) and deleterious fitness score (bottom gray line). **c** Residue-level mean growth-associated fitness heatmap on the target structure, with increasing fitness represented as a gradient from red to blue. **d** Residues with high mean growth-associated fitness (green sticks), and highly conserved and functionally important Bridge Helix (BH) residues (orange sticks). (Left) BH (bridge helix) residues with charged side chains (blue sticks) and proximal target residues with high growth-associated fitness means (red sticks). **e** (Right) A bar plot (blue) of mean growth-associated fitness for substitutions to amino acids with positively charged (blue) and negatively charged (red) sidechains respectively. Within each box, the horizontal black lines represent median values, lower and upper bounds

correspond to the 25th and 75th percentile, and the whiskers extend to the extreme values within the 1.5× interquartile range. The top and bottom dashed lines represent the cutoff for beneficial and deleterious fitness scores respectively. There were 6, 2, 2, 28, 34, 10, and 4 independent observations for each variant left to right in the plot. An observation represents an independent fitness measurement of a synonymous variant of the focal mutation in two biological replicates (Supplementary Note 1). **f** Correlation of growth-associated fitness with the enrichment for CR703 resistance for target mutations. The dashed vertical and horizontal lines represent the cut-off for beneficial growth-associated fitness and CBR703 resistance respectively. **g** Residues with greater than five CBR703 resistant mutations (all spheres) and ones with high growth-associated mean fitness (green spheres). **h** Residues with high growth-associated fitness (green sticks) with red lines connecting two residues with significant positive epistatic interaction (Supplementary Fig. 4 and Supplementary Note 2). Source data are provided as a Source Data file.

Of 138 double mutants with both high growth and the stringent phenotype, 46% (64) variants were a combination of stringent and growth-improving mutations (Blue, Fig. 6c). Additionally, 76.1% (105) variants occurred in positions where one position had high mean growth-associated fitness and the other had high mean stringent enrichment (Pink and Blue, Fig. 6c). Growth-stringent double mutants had both higher growth-associated fitness (KWH-test, $p$-value < $10^{-16}$) and stringent enrichment (KWH-test, $p$-value < $10^{-16}$) compared to combined synonymous mutations (Fig. 6d). Therefore, residues clusters determining the growth and the stringent

phenotypes were modular, and combining growth-improving and stringent mutations led to both phenotypes.

**Residues associated with each phenotype are highly conserved**
The growth-improving and the stringent phenotypes were each associated with a partial loss of an interaction. The target region is highly conserved, naturally. Therefore, each interaction may have evolved to have functional importance. Particularly, the stringent phenotype was associated with polar interactions in loops, and the presence of polar interactions in (usually flexible) loops signifies

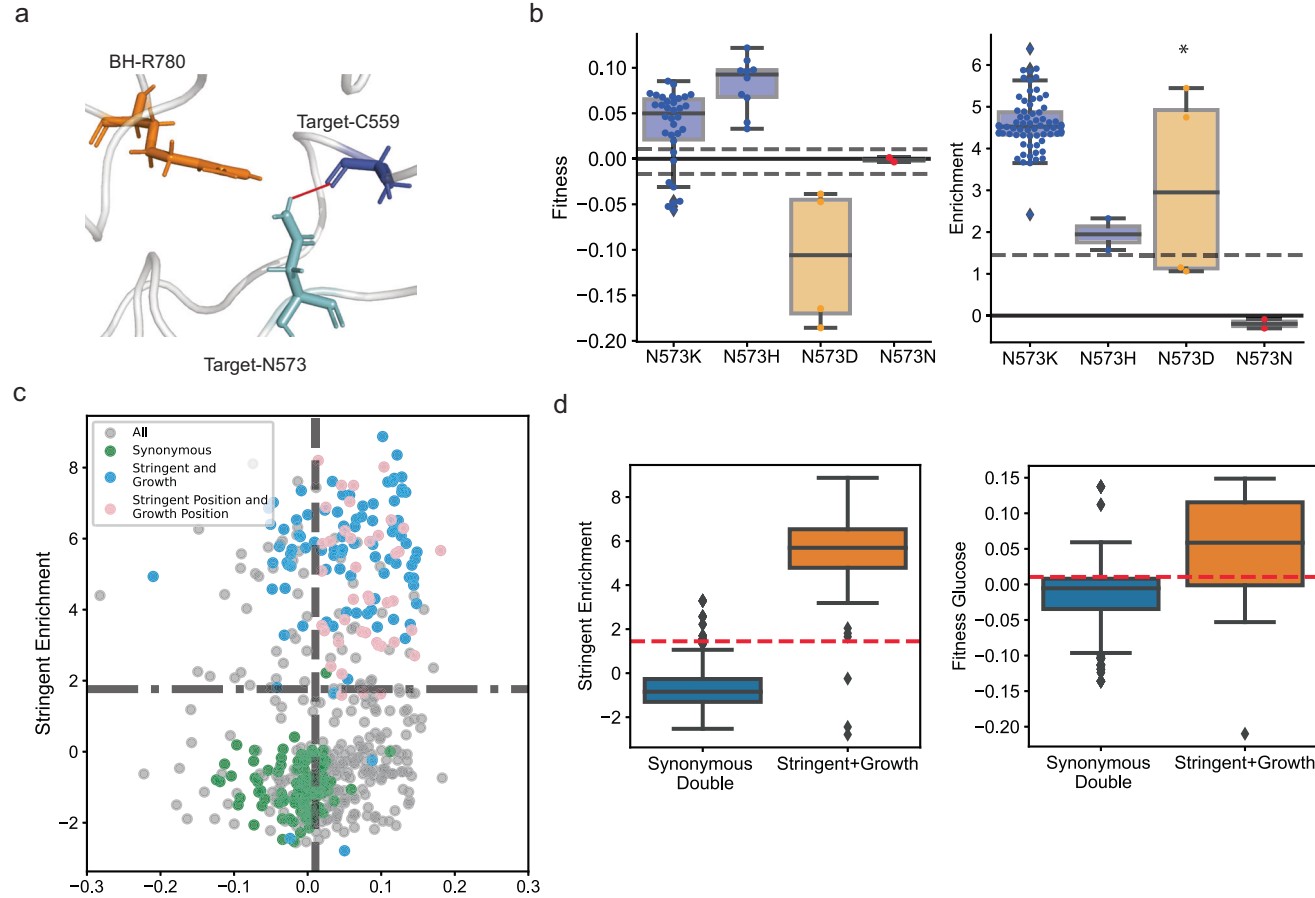

**Fig. 6 | Some mutations are both growth-improving and stringent. a** The residue N573 (cyan sticks) with internal polar interaction (red line) with residue C559 is close to BH positively charged residue R780. **b** A bar plot (blue) of mean growth-associated fitness for substitutions to amino acids with positively charged (blue), negatively charged (red), and synonymous substitutions (red). On the left, the top and bottom gray lines represent the cut-off for beneficial fitness score and deleterious fitness score respectively. On the right, the dashed line represents cut-off for stringent mutations. Within each box, the horizontal black lines represent median values, lower and upper bounds correspond to the 25th and 75th percentile, and the whiskers extend to the extreme values within the 1.5× interquartile range. There were 34, 10, 4, and 2 independent observations for Glucose-associated fitness (left) and 72, 2, 4, and 2 independent observations for stringent enrichment

(right) for each variant left to right in the plot. An observation represents an independent fitness measurement of a synonymous variant of the focal mutation in two biological replicates (Supplementary Note 1). **c** Correlation of growth-associated fitness and stringent enrichment of all double mutations (grey), and ones with combined synonymous mutations (green), combined growth-improving and stringent mutations (64, blue), and combined mutations on positions important for growth and stringent phenotypes (41, pink. 105 total). **d** Box plots comparing stringent enrichment (left) and growth-associated fitness (right) of double synonymous mutations (136 variants, blue) and double growth-stringent mutations (91 variants, orange). The red line represents cut-off for stringent mutations (left) and the cut-off for beneficial growth-associated mutations (right). Source data are provided as a Source Data file.

functional importance. We looked at conservation at different levels of divergence, i.e., an alignment of RpoB sequences from proteo-bacteria, and bacteria, and against representative sequences from eukaryotes. Residues with high mean growth-associated fitness (12 of 13 residues with <5% variability), and high mean stringent enrichment (16 of 17 residues with <5% variability) were highly conserved in proteobacteria (Fig. 7a). In bacteria, while higher variability was observed in growth-associated residues (8 of 13 residues with <5% variability), residues with high mean stringent enrichment remained highly conserved (15 of 17 residues with <5% variability). ppGpp-mediated Stringent response is also observed in plant chloroplasts[45]. We observed that 15 of 17 residues were highly conserved even in plant chloroplast RNAP. The stringent response is absent in archaea and eukaryotes. In both eukaryotes and archaea, the stringent phenotype-determining DNA-proximal motif between residues 531–538 was absent. Therefore, we observed significant conservation of growth-associated residues in proteobacteria and stringent residues in stringent response-associated species across kingdoms in bacteria and plant chloroplasts.

## Partition of residues into modules allows compensatory evolution

The two variable residues with high mean stringent enrichment were T553 and N573 (Fig. 7a). However, in the case of T553, the only observed substitution was T553S, which would still retain the polar interaction. The other variable residue was N573, where mutations lead to the combined growth-stringent phenotype (Fig. 6a, b). This variability suggests that having growth and stringent traits partitioned into independent residues and interactions may provide an advantage of compensatory evolution. If a mutation in one cluster improved a trait at the cost of the other, a second mutation in the second cluster may compensate for the cost (Fig. 6). The compensation likely provided greater success for mutations in residue N573 compared to mutations optimizing only one phenotype.

We looked at Rifampicin resistance mutations to test the compensation hypothesis. Rifampicin is an essential drug against tuberculosis[20]. Several stringent mutations in the target also confer resistance to Rifampicin and have fitness costs[20]. Mutations within the RNAP have compensated for the fitness defects and allowed resistant

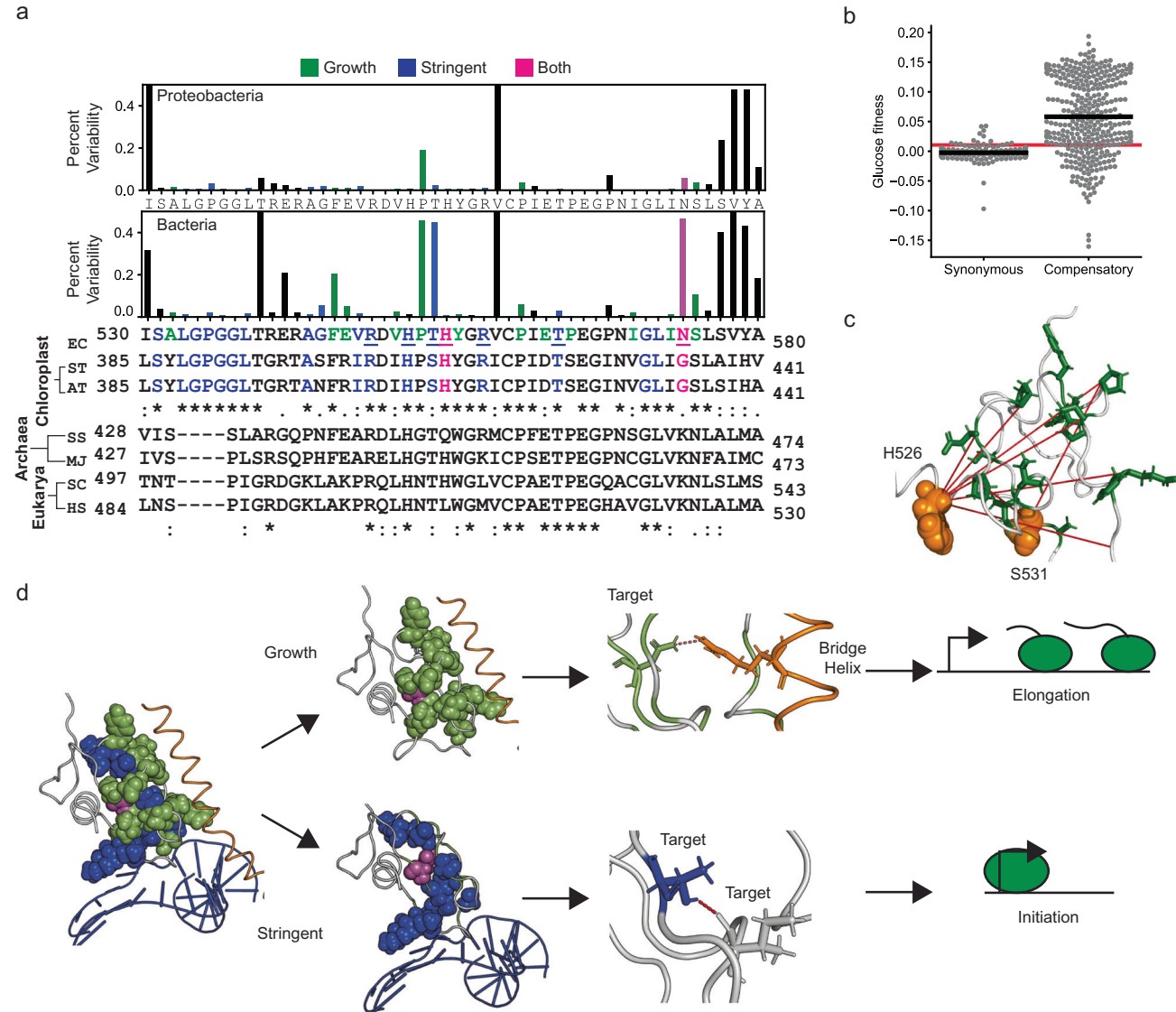

**Fig. 7 | Conservation and compensatory evolution. a** Bar plot of percentage variability in residues with high growth-associated fitness means (green), high mean stringent enrichment (blue), and growth-stringent residues (pink). (Bottom) An alignment of the target sequence from *E. coli* with the RNAP IV sequence from plant chloroplasts from *Sansevieria trifsciate* (ST), *Arabidopsis thaliana* (AT), RPB2 subunit of RNA polymerase II from *Homo sapiens* (HS), *Saccharomyces cerevisiae* (SC) and RPO1N RNAP subunit from *Methanocaldococcus jannaschii* (MJ) and *Saccharolobus solfataricus* (SS). Source data are provided as a Source Data file. **b** Distribution of growth-associated fitness for synonymous mutations and compensatory mutations for Rifampicin resistance. The statistics were derived using n-synonymous = 186 and n-Compensatory = 356 measurements of Glucose fitness. The red line represents the cutoff for beneficial growth-associated mutations. **c** Residues with high growth-associated fitness (green sticks) with significant positive epistasis (red lines) with rifampicin-resistant stringent mutations (orange spheres). **d** Summary: two modular clusters in the target region associated with unique interactions and possibly functions control the growth and stringent phenotypes.

strains to propagate and exacerbate the Rifampicin resistance crisis[20–22]. In a previous study, we observed significant positive epistasis amongst the double mutants in the target, with Rifampicin-resistant mutations combined with neutral mutations[32]. We extracted this list of compensatory non-resistant target mutations from the previous study. We found that the compensatory mutations had significantly higher growth-associated fitness compared to synonymous mutations (KWH-test, *p*-value < 10$^{-16}$, Fig. 7b). Nearly 76% (271 of 356 total mutations) of the mutations had high growth-associated fitness mean (Fig. 7b). Rifampicin resistance mutations in residues H526 and S531, which account for the majority of clinically known mutations in *Mycobacterium tuberculosis*[46], show significant positive epistasis with multiple growth-improving residues within the target (Fig. 7c). Therefore, the partitioning of functions in different clusters enables compensatory evolution.

## Discussion

Mutations in the RNAP provide access to complex phenotypes and are frequently selected in adaptive laboratory evolution to diverse stresses. RNAP mutations are highly pleiotropic[3,26], which made it difficult to identify the traits under selection during adaptive evolution. The current dominant hypothesis is that RNAP may rewire transcription for condition-specific adaptation. However, within the scanned area, the fitness of the beneficial RNAP mutations strongly correlated between five diverse environments (Fig. 2). Not only were the same mutations adaptive in multiple conditions, but the degree of adaptation for a beneficial variant between conditions was also comparable (Fig. 2). Therefore, a common underlying mechanism improved fitness across conditions. The only commonality between the conditions was base M9-media, and all reconstructed mutations had a higher growth rate in M9 minimal media compared to wild-type (Supplementary Fig. 2).

Therefore, during laboratory evolution selection may favor RNAP mutations that impact global traits such as growth as opposed to previously proposed condition-specific adaptation (Fig. 2). Similar cross-environment adaptation has been reported previously for other RNAP mutations[3,10,26]. Mutations within the region found during ALE in other conditions also had high growth-associated fitness in our analyses (Supplementary Table 1). This suggests that the selection for improved growth extends beyond our tested conditions.

Growth control is intricately tied to stringent regulation in *E. coli*. Growth-improving mutations had a delayed stringent response (Supplementary Fig. 2c, d). During the stringent response following an increase in the levels of ppGpp, the binding of ppGpp and dksA to the RNAP activates the expression of starvation/maintenance-associated genes such as the ones involved in amino acid biosynthesis. By imposing starvation for amino acids in *ΔrelAΔspoT* strain, we also found multiple stringent mutations of the RNAP, that favor amino acid biosynthesis with the trade-off of reduced growth (Fig. 2). The stringent mutations mimic the ppGpp-bound state of the RNAP. Therefore, they may also activate the expression of other maintenance functions to again alter global traits. In line with our proposition that RNAP mutations select for altered global traits, it is important to note here that, stringent mutations of the RNAP have also been selected for maintenance-associated adaptation to stresses such as long-term starvation, tolerance, and resistance to antibiotics and chemicals[12–14].

We found that the global traits of growth and maintenance were controlled by two modular residue clusters, each associated with the loss of a specific molecular interaction (Figs. 4,5, and 7d). The coupling of the improved fitness with a loss of molecular interactions led us to an atypical distribution of fitness effects, with a prominent mode of beneficial mutations (Fig. 5a). The higher fraction of beneficial mutations occurs as molecular interactions can be decreased in multiple ways (as mutations are more likely to decrease than increase function), as opposed to the rare beneficial mutations observed in previous protein DFE[40] that resort from rare gain-of-function.

What are the involved molecular interactions? The stringent phenotype is known to be determined by the stability of an open complex intermediate during initiation, controlled by target-DNA interactions[28,39]. Accordingly, modifications in DNA-proximal residues (within 10 Angstroms of the target) and internal pairwise interactions (likely associated with open complex stability) were stringent (Fig. 4). The growth-associated fitness was linked to a decreased interaction between the target-BH (Fig. 5). The decreased target-BH interaction increases the transcription elongation speed and pause resistance during elongation[42]. Mutations in other regions of the RNAP known to increase elongation and pause resistance also improved growth (Supplementary Fig. 6). Similarly, growth-improving mutations in other regions of the RNAP have also been found to have an increased elongation rate and pause resistance[42,47–49]. Therefore, an increase in catalysis and pause resistance increases the growth rate. An increase in elongation speed may increase the concentration of free RNAP, which is known to favor growth[50]. Additionally, transcriptional pauses are known to limit elongation for growth-associated genes[51]. We cannot completely exclude the possibility that the target-DNA and target-BH interaction may alter multiple other steps during transcription[49]. However, the functions controlled by the target-DNA/internal and target-BH interactions have to be modular because upon combining the loss of internal interaction and reduced target-BH interaction we obtain variants with both an improved growth and the stringent phenotype (Fig. 6). If both interactions affected overlapping functions, their independent optimization would not be possible.

It has always been thought that cells cannot improve both growth and stringent regulation due to molecular-level and systems-level constraints. At the systems level, it is proposed that the stringent response controls the allocation of the "limited" resources between growth and maintenance[7] and a reallocation of the limited resources

causes tradeoffs[3]. However, the ability of cells to both improve growth and have the stringent phenotype shows that resources may not be limited between the two objectives at a systems level. Interestingly other adaptive laboratory evolution experiments have led to evolved *E. coli* that break growth-survival tradeoffs as well[52]. Several studies have reported the presence of excess resource reserves in exponentially growing cells[53,54]. However, if resources are not limited, why would the improvement of one trait cause tradeoffs with the other? The tradeoff is likely molecular. It is well known that the decreased open complex intermediate stability of stringent mutations decreases transcription initiation for growth-determining ribosome biosynthesis genes[28]. Additionally, an increased transcription rate may increase free RNAP availability, which is known to inhibit maintenance-associated genes such as amino acid biosynthesis[50].

The combined growth-stringent phenotype showed that the growth-improving mutations can overcome the tradeoffs associated with the stringent mutations and vice versa. Therefore, at a molecular level, an increase in elongation rate may compensate for the decrease in initiation for ribosome biosynthesis genes. In confirmation of this hypothesis, in vitro studies indeed show that increased elongation speed alters DNA supercoiling to increase open complex stability and compensates for initiation defects of stringent mutations[28]. Evidence of other in vitro feedback mechanisms between elongation and transcription initiation also exists[55].

Multiple clinical examples of the cell's ability to overcome tradeoffs further demonstrate that multiple cellular traits can be optimized independently[2]. Compensatory evolution is observed for growth-impacting antibiotic resistance mutations and even observed in cancer. Stringent mutations at position 531 confer Rifampicin resistance and cause growth defects[56]. The cost of resistance should prevent resistance propagation. However, mutations within the RNAP can compensate for the fitness defects to allow the propagation of resistant strains[20] and even promote the emergence of extremely drug-resistant and multidrug-resistant strains[20,21]. We demonstrate that the partition of the traits in different clusters provides the evolutionary advantage of compensation (Fig. 7). The growth-improving mutations of the RNAP compensate for Rifampicin resistance (Fig. 7). The control of the tradeoff-associated traits by modular handles enables evolutionary plasticity. The modularity of function may extend to the entire RNAP because compensatory interactions are observed even between RNAP subunits[20]. Such "modules controlling different traits" could also be relevant in other biological systems such as cancer where growth and maintenance tradeoffs are also challenged[57].

Within the scanned region of the RNA polymerase, we identified modules for global growth and maintenance control, that may be associated with transcription elongation and initiation respectively. However, the RNA polymerase complex also undergoes environment-specific conformational and functional changes to alter transcription. While mutations in other regions of the RNA polymerase affecting elongation speed/pausing and open initiation complex stability may affect growth and maintenance (Supplementary Fig. 6), a condition-specific adaptation via mutations in other regions of the RNA polymerase is also possible. Deep mutational scanning of the RNA polymerase in multiple environments and genetic backgrounds using technologies such as CREPE provides a strong platform to discover such residues and residue modules (Fig. 1a). In addition, the discovery of such modules opens lucrative avenues for predictable cellular control; with significant applications in drug design, to prevent propagation of antibiotic resistance, and enable strain engineering in biotechnology and synthetic biology. RNAP is a top target for antibiotics. We can design effective drugs and drug combinations to target the two interactions to simultaneously inhibit multiple vital global traits i.e., growth and maintenance. In the case of Rifampicin resistance, fitness defects are likely compensated by improving growth, determined by a decreased target-BH interaction (and possibly

increasing elongation rate and pause resistance). Therefore, we can design drugs that strengthen target-BH interaction or alter transcript elongation to overcome compensation.

Despite frequent variation in the laboratory, both the growth and stringent phenotype-determining residues are highly conserved (Fig. 7). Residues with high mean stringent enrichment are conserved even across kingdoms i.e., in plant chloroplasts. Each residue is also associated with an interaction and in each case the cells improved fitness by losing the interactions. As opposed to the controlled laboratory evolution, bacteria such as *E. coli* have naturally evolved to survive in fluctuating environments such as in cycles of feast and famine[58]. Most growth-favoring and stringent mutations were specialists for either growth or stringency and were associated with a tradeoff with respect to the other objective (Fig. 2). Such specialist mutations with one objective optimized at the cost of others may be counter-selected in fluctuating environments[2]. While the counterselection of specialists explains the residue conservation of most residues, it does not explain the conservation (specifically within proteobacteria) of residues such as N573 where mutations optimized both objectives.

The growth-determining and stringent residues and their associated interactions likely have functional importance, as residue conservation is also a strong predictor of functional importance[35]. Since, each phenotype is associated with a loss-of-function, it is likely that the functions associated with each residue cluster is important for survival in the wild. The residues may be important for regulation. In addition to performing transcription, the RNAP also regulates global transcription. Regulation requires a change (activation or repression) of gene expression in response to a change in the environment. However, the maintenance functions such as amino acid biosynthesis are constitutively activated in the stringent variants. Therefore, the stringent mutations lose the ability to regulate gene expression in response to starvation, suggesting the regulatory role of the stringent residues. Similarly, the target-BH interaction may regulate growth. The small molecule ppGpp is known to regulate transcription elongation for growth-associated ribosome biosynthesis genes by increasing pausing[59,60]. The pause-resistance of the growth-promoting mutations likely bypass these regulatory pauses to promote growth. Therefore, we propose the described residues and the associated interactions may have a regulatory function. The dissonance between residue conservation and laboratory-determined fitness suggests that the associated function is not important in the controlled selection environment and even provides a significant fitness advantage. Similar loss-of-regulation has been previously selected in multiple laboratory evolution experiments[61–63]. However, further biochemical, transcriptomic, and computational characterization of the mutations is required to understand the functional/regulatory role of the residues. We also need to characterize the mutations in a common background because the *ΔrelAΔspoT* background may introduce unpredictable global changes.

Our observations show a crucial dichotomy of adaptive laboratory evolution. On one hand, the controlled environment can select for a loss of function, putting into question the ecological relevance of mutations identified in adaptive laboratory evolution. However, on the other hand, combining adaptive evolution with high-throughput approaches such as ours, can in turn help us discover these interactions and functional modules. Beyond our analysis of the RNAP, the CREPE-based approach can be used to study other complex protein systems. High-throughput fitness estimates in the native genomic context allow us to successfully extrapolate the biochemical understanding of complex proteins to a systems understanding of phenotypic traits to build sequence-cellular fitness maps.

## Methods
### Media recipe
All cultures for genome editing experiments were performed in Lysogeny broth (LB). All adaptation experiments were performed in M9 minimal media prepared by adding 20 mL 5X M9 salts (BD 248510), 2 mL Glucose (20%), 200 μL MgSO₄, 10 μL CaCl₂, and Thiamine to a final concentration of 0.01% in 98 mL water. In the selection with different sugars, 20% Galactose and 20% Glycerol were used. For selection with different stresses, we added NaCl to a final concentration of 300 mM for high osmolarity and Butanol to a final concentration of (0.6% v/v). For fluorescence measurements for induction of stringent response, the media was supplemented with 0.2% Cas amino acids.

### Strains, plasmids, and cloning methods
All evolution experiments and mutant validation were done in *Escherichia coli* strain MG1655 substrain K12. The *ΔrelAΔspoT* library was prepared using the strain MG1655 CHN 188: rph-1 ΔrelA::FRT ΔspoT::Kan. We worked with two *ΔtolC* strains tolC::Tn10 and ΔtolC::Kan from the KEIO collection[64]. We used the gRNA and repair template described previously[44]. We used the pCREPE plasmid with the *cas9*+lambda Red recombination+*mutL-E32K* integrated into the same plasmid backbone from our previous study for Cas9-mediated recombineering. The gRNA plasmid, where the gRNA was expressed under the J23119 promoter, was purchased from Addgene (https://www.addgene.org/71656/), which was a gift from Dr. Ryan Gill's lab. We cloned the spacer (Supplementary Table 5) into the plasmid for the Cas9-mediated recombineering. We constructed the template for the error-prone PCR libraries in the pSAH031 backbone (https://www.addgene.org/90330/), which was a gift from the Dr. David Savage's lab[65]. We cloned the target region for the error-prone PCR using CPEC assembly. We amplified the target region with 250 base pairs of end homologies in the pSAH031 backbone using the primers F_rpoB_W1 and R_rpoB_W1 (Supplementary Table 5) and the backbone using F_pSAH_rpoB and R_pSAH_rpoB (Supplementary Table 5) using the Kapa Biosystems high-fidelity polymerase (catalog #07958897001). Subsequently, we replaced the wild-type sequence in the error-prone plasmid with 250-bp-long gblocks (Eurofins) with the synonymous PAM mutation (SPM) using CPEC cloning (Quan & Tian, 2011).

For CPEC cloning, we used 12.5 μl (with at least 100 ng of the backbone) of an equimolar insert: backbone mixture, and 12.5 μl of NEB 2× Phusion Master Mix (catalog #M0530). We used the following PCR protocol: 98 °C-30 s, 10× (98 °C-10 s, 65 °C-10 s, 72 °C-90 s) and a final extension at 72 °C for 120 s followed by a hold at 12 °C. All amplifications were performed using Kapa Biosystems high-fidelity polymerase (catalog #07958897001). 10 μl of the CPEC reaction was dialyzed using a 0.45-micron dialysis membrane. The dialyzed reaction mixture was transformed into commercial Top10 competent cells. After 1 h of recovery, the cells were plated on LB agar plates with the appropriate antibiotics.

### Error prone PCR and construction of repair template
We constructed the error-prone PCR libraries using the Agilent GeneMorph II Random Mutagenesis Kit (Part #200550). We used the primers RpoB1_mut_f and RpoB1_mut_r to amplify the target region (Supplementary Table 5). For the PCR we used 10 ng of the template plasmid. We then amplified the target region using the following protocol: 95 °C-2 min, 30× (95 °C-30 s, Tm-30 s, 72 °C-1 min), and final extension: 72 °C-1 min. Following the error-prone PCR, 1–2 μl of NEB's DpnI (Catalog # R0176L) was added directly to the PCR, and the reaction mix was incubated for 2 h at 37 °C. The error-prone PCR template was purified using Qiagen's gel purification kit (Catalog #28704). The purified PCR product was cloned into the pSAH03 backbone with the flanking 250 bp homology arms using the NEBuilder HiFi DNA assembly kit (catalog #E2621) using the manufacturer guidelines for reaction setup. The NEBBuilder assembly reaction mix was then transformed into Lucigen Elite E. cloni electrocompetent cells (catalog #60061) and plate several dilutions of the transformation reaction on LB agar + kanamycin (the antibiotic marker for pSAH03). After overnight growth at 37 °C, 50,000–100,000 colonies were

collected by scraping the plates in liquid LB. The plasmid-error-prone PCR library was then extracted using the Qiagen Miniprep extraction kit (Catalog 27104) following the instruction manual. The donor library repair template was amplified using 10 ng of the plasmid template and the primers F_rpoB_W1 and R_rpoB_W1 (Supplementary Table 5). We performed the PCR using the KAPA HiFi polymerase (catalog #07958897001) following the Manufacturer's guidelines. We amplified the library using the following protocol: 95 °C-2 min, 15× (98 °C-20 s, Tm-15 s, 72 °C-1 min), and final extension: 72 °C-5 min. To avoid over-amplification, we performed only 15 PCR cycles. After the PCR, we used agarose gel extraction and purification using the Qiagen gel extraction kit following the manufacturer's protocol.

### Cas9-mediated recombineering
Cas9-mediated recombineering was used to introduce genomic mutation following the heat-shock protocol[66]. Desired *E. coli* strains with the pCREPE plasmid were grown at 30 °C overnight. In the morning, overnight cultures were diluted 100-fold into fresh media. The cultures were grown at 30 °C until mid-log optical density (measured at 600 nm) of 0.4–0.5. The cells were placed in a shaking water bath set at 42 °C to induce the lambda Red recombination operon and *mutlL-E532K* gene. Then the heat-shocked cells were placed on ice immediately and chilled for 15 min. The chilled cells were centrifuged at $7500 \times g$ at 4 °C for 3 min. The pellet was washed with 25 ml ice-cold 10% glycerol solution by resuspending and centrifuging at $7500 \times g$ at 4 °C for 3 min thrice. After the washes, the cells were finally resuspended in ice-cold 10% glycerol concentrated to 150-fold compared to the starter culture volume (-250 uL for a 50 mL culture). Dialyzed mixtures of the repair template with the gRNA were electroporated into 50 µl of the competent cells at 1.8 kV. The cells were finally recovered for 3 h at 30 °C in 1 mL LB and subsequently plated on LB agar plates with chloramphenicol and 100 ug/mL spectinomycin (the resistance marker for the gRNA plasmid). The plates were grown overnight at 30 °C. Subsequently, ~50,000 colonies were scraped and stored as glycerol stocks for the subsequent selection experiments.

### Selection experiment in different liquid media
Adaptation in minimal media with different sugars and stresses was performed using serial dilution. Two independent glycerol stocks (to make independent biological replicates) with the *rpoB* libraries were thawed and a preculture was started by diluting 1 mL of the glycerol stock in 50 mL M9 Glucose minimal media for 4–5 h. Bottles with 50 mL of media with different sugars and stresses were preheated at 37 °C. When the OD of the preculture reached an OD of 0.2, we inoculated two bottles with each sugar or stress condition with an independent preculture to a 32-fold dilution. Then we followed the growth of each culture (Supplementary Fig. 4). When the cells reached an OD of 0.2, we re-diluted the cells 32-fold in fresh cultures. So, between each serial dilution, we had 5 generations of growth. The serial dilution was performed for 30 generations, with dilution and sampling at every 5 generations (Supplementary Fig. 4).

### Selection for the stringent phenotype and CBR703-resistant mutants
We constructed the RpoB library in the *ΔrelAΔspoT* and the *KeioΔrelA* libraries respectively. Before selection, three glycerol stocks for each library were thawed. The three glycerol stocks were mixed and recovered in LB at 37 °C for four hours. After the recovery, several dilutions for each library were plated on agar plates with their respective selection media. The stringent mutants were selected on M9 minimal media plates supplemented with Glucose. Selection for CBR703 mutants was performed on LB plates with 8 ug/mL of CBR703. A 100 uL sample of the cells was boiled before selection for sequencing. Around 10,000 colonies were scraped after selection and glycerol stocks were saved for subsequent analysis.

### Next-Generation sequencing
DNA for next-generation sequencing was extracted using the boiling protocol. 50 uL of the cell sample for each condition was washed twice with PCB. For washing the cells were centrifuged at 7500 g and then resuspended in 1 mL PBS. The resuspended pellet was boiled at 100 °C for 10 min and placed on ice immediately. Subsequently, the target region was amplified for sequencing, and to attach the next-generation sequencing adapters using the RpoB_nextgenseq_for and RpoB_next-genseq_rev primers (Supplementary Table 5). The PCR was performed using KAPA HiFi polymerase with a reaction mix containing 5 uL of the cell extract, 0.25 uL of 100 mM primers, 25 uL of KAPA polymerase, and water to make the reaction up to 50 uL. We amplified the library using the following protocol: 95 °C-2 min, 15× (98 °C-20 s, Tm-15 s, 72 °C-1 min), and final extension: 72 °C-5 min. To avoid overamplification, we performed only 15 PCR cycles. The sequencing was performed using paired-end 2 × 150 np sequencing on the Illumina NextSeq platform.

### Growth curves
Growth curves were determined using a high-precision technique developed in-house and using a microplate reader. For the in-house machine, we used a lab-made tubidometer with quasicontinuous and parallel measurements for cultures in 15 ml glass culture tubes. The overnight cultures were diluted 1/1000 in 5 mL cultures and OD was measured at 30 °C at 200 rpm using a phototransistor every 10 s for a range of emitter light intensities. Appropriate intensities were used to determine the growth curves[67].

To estimate growth rates for reconstructed strains, overnight cultures for each mutant in M9 minimal media were diluted 1/1000 and dispensed in 96-well plates for a volume of 200 uL of media for the growth estimate. The OD600 was measured every at 37 °C with shaking over 24 h using the Tecan microplate reader.

### Fluorescence measurements
To estimate growth rates for reconstructed strains, overnight cultures for each mutant in M9 minimal media were diluted 1/1000 and dispensed in 96-well plates for a volume of 200 uL. The OD600 and fluorescence was measured every 2 min at 37 °C with shaking over 15 h using the Tecan Infinite M200 PRO microplate reader. For GFP-based reporters, we used an excitation wavelength of 480 nm and an emission wavelength of 510 nm.

### Data analysis
Within the code, we used custom packages: Python 3.7 and packages within Python SciPy, SciKitLearn, Numpy, and Pandas.

**Preprocessing.** All analysis was performed using a custom analysis pipeline for CREPE[44]. Paired-end reads were merged using the Usearch Mergepairs algorithm[68]. Then, the assembled reads were aligned with the wild-type sequence as the reference using the usearch_global tool from the usearch package[68]. The alignment generated a text file with a detailed output, from which the alignment was used to identify the mismatches. Then using the unique mismatches/sequences, we obtained the value counts for each variant. We then used a custom code to extract the amino acid changes[32].

**Clustering and filtering of data.** Often, mutations are introduced during the PCR steps of library preparation and during the sequencing itself. Therefore, several mutations detected in the dataset are not variants present during the selection but an artifact of errors introduced. The frequency of such errors is low and the sequences associated with such errors can be clustered with their possible parent sequence. As a control, we sequenced regions around the target which were not mutated. We determined the average error rate associated with the sequencing prep using these regions of the sequence. Subsequently, we wrote a custom

algorithm to cluster the possible error-associated sequences with a parent.

We further filtered the data to eliminate reads that may be erroneous. Since we targeted an essential gene RpoB, the occurrence of stop codons would be impossible. Therefore, the stop codons provided another way of identifying the frequency of erroneous sequences. We made the distribution of frequency associated with sequences with stop codons. Only sequences with frequencies above the 99th percentile of this distribution were used for all analyses.

**Estimation of fitness for liquid cultures.** The fitness scores were estimated using a previously described algorithm by Rubin et al., 2017[69]. Briefly, the count of each variant (i) was divided by the count of the wild type (wt) to get a frequency ratio at a time point t:

$$p_{i,t} = \frac{(c_i + 0.5)}{(c_{wt,t} + 0.5)} \tag{1}$$

where c designates the counts. Then we regress in $M_{i,t}$:

$$M_{i,t} = \log(p_{i,t}) \tag{2}$$

We performed weighted linear least squares regression with the weight $(V_{i,t})^{-1}$;

$$V_{i,t} = 1/(c_{i,t} + 0.5) + 1/(c_{wt,t} + 0.5) \tag{3}$$

Fitness was estimated as the slope of the regression line.

**Enrichment estimates.** The enrichment for *ΔrelAΔspoT and the KeioΔrelA* libraries was estimated using our previously built pipeline (https://github.com/Alaksh/CREPE-Analysis-Code). The scores were calculated using previously described algorithms by Rubin et al.[69] for fitness estimates with two-time points (pre-and post-plating on selection media):

$$\text{fitness}, f = \log\left(\frac{(c_{i,sel} + 0.5)}{(c_{wt,sel} + 0.5)}\right) - \log\left(\frac{(c_{i,input} + 0.5)}{(c_{wt,input} + 0.5)}\right) \tag{4}$$

where $C_i$ is the total count for a variant "i" in the library and $C_{wt}$ is the total count for the wild-type reference in the library. "Sel" signifies the counts obtained after selection on rifampicin, and "input" signifies counts before selection. For each score, we estimated an error using a Poisson approximation.

$$\text{Standard error} = \text{sqrt}\left(\frac{1}{c_{i,sel}} + \frac{1}{c_{wt,sel}} + \frac{1}{c_{i,input}} + \frac{1}{c_{i,input}}\right) \tag{5}$$

### Structural analysis
All structural analysis was performed using version 4.6.0 – Build 27.20.100.9664. We used the PDB model 6B6H[70] for all structural analysis, and structure 4ZH2[43] was used for CBR703 bound structures.

### Reporting summary
Further information on research design is available in the Nature Portfolio Reporting Summary linked to this article.

## Data availability
All sequencing data has been deposited at the Europen Nucletide Archive (ENA) and are accessible through the project accession number PRJEB59215. The sequencing data can also be accessed at DRYAD database: https://datadryad.org/stash/share/pMgkxJauRphu1eU49Yyx LOznG6zakpHaWBgLqzCcL-Y. The processed data is accessible with the Dryand link: https://datadryad.org/stash/share/ZfUDt7kbGXPGw4_ tl_roBQ2okJde3oKFftqgAwYlF0o. Source data is provided with the paper. Source data are provided with this paper.

## Code availability
The code used to analyze the data is available at Zenodo (https://zenodo.org/record/80340940)[71].

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

## Acknowledgements

We would like to acknowledge the University of Colorado Boulder's next-generation sequencing core headed by Dr. Amber S. Scott at the University of Colorado, and Melanie Magnan at INSERM for support with deep sequencing for our study. We would like to thank Dr. Erick Denamur, Dr. Ivan Matic, Dr. Philippe Nghe, and Maureen Micaletto for useful discussions and for proofreading our manuscript. A.C. and R.T.G. were funded by the US Department of Energy grant DE-SC0018368 awarded to R.T.G. A.C., B.G., Z.D., R.F., and O.T. were funded by grants European Research Council, FP7 grant 310944, the French Agence Nationale pour la Recherche ANR GeWiEp (ANR-18-CE35-0005-01), the Fondation pour la Recherche Médicale (EQU201903007848). A.C. was also funded by the 2017–2018 STEM Chateaubriand Fellowship sponsored by the French Institut National de la Santé et la Recherche Médicale (INSERM).

## Author contributions

A.C., R.T.G., and O.T. conceived the study. A.C. and O.T. designed all the experiments. A.C. performed the selection experiments, sequencing library prep, sequencing, structure, and data analysis. B.G., and Z.D. built and characterized the growth of the individual reconstructed mutations. R.F. helped with the sequencing data analysis. A.C. and O.T. wrote the paper with input from all authors.

## Competing interests

The authors declare no competing interests.
