## [Peer Review File · Nature Communications]

Deep mutational scanning reveals the molecular determinants of RNA polymerase-mediated adaptation and tradeoffsREVIEWER COMMENTS

Reviewer #1 (Remarks to the Author):

The topic of this paper is RNAP mutations that are acquired during evolution and studying their causality. The origin of these studies was the Herring et al paper in Nature Genetics in 2006, where for the first time rpoC mutations from laboratory evolutions were introduced into the starting strain showing their clear causality. Balas Papp et al reviewed this story in a Nature Reviews Genetics paper (I forget the year), and ref 4 did a multiscale study on the mechanisms of causality, except at the structural level where the current study makes a notable contribution. The primary methods used here is to create a library of mutants using targeted mutagenesis and screening the populations of mutants using enrichment cultures. The paper and its results are suitable for Nat Comm, but there are several issues that need to be addressed. It is a very good paper, well written. Few minor grammatical errors.

Larger issues

How can cells have both stringent response and growth-promoting mutations simultaneously? My understanding is that these are two opposite forces, and it is impossible to both famine and feast at the same time. If this were possible, it is unclear why E. coli RNAP wouldn't already have said mutations.

A large amount of the work is to demonstrate how variants with substantial phenotypes cluster near or far from each other. However, the figures to demonstrate this are largely qualitative. It would be good to have some more quantitative analysis, like Grantham scores, changes in polarity, or some structural predictions.

What is the molecular basis of increased stringent fitness? Do stringent genes have different promoter sequences than growth-promoting genes? Is it about interaction with other trans-acting proteins? Why did the increased catalytic rate and pause resistance by growth-promoting mutations not increase expression of stringent genes?

The argument that enriched mutations found in a relA and spoT KO'd strain are evidence that stringent-promoting mutations exist in RNAP is a stretch. Removing such a large regulatory element will have very large effects on the transcriptome that will be difficult to detangle. Did you try any other methods such as imposing amino acid starvation on wildtype E. coli?

A primary concern is that the authors work to demonstrate how variants with substantial phenotypes either cluster near each other or far away from each other. However, the main figures meant to demonstrate this are qualitative (examples: Figure 3B, Figure 4E), and there are better approaches that allow for a more quantitative representation. This more quantitative representation of the distance between variants and their potential clustering would be accomplished through analytics.

Amino acid substitutions may have different functional/structural consequences depending on the similarities between the original amino acid and the substituting residue. Thus, Fig 4A and Fig 5B may be misleading. That is: Some positions may seem to be not stringent, if there's only substitutions that do not change much (Leucine to isoleucine i.e), although the position is stringent. It would be great if the authors incorporate some kind of measure to estimate the consequence of the substitution (Grantham score maybe?).

Smaller issues

Define certain terms - close proximity for example, how many Angstroms is that?

Clarify certain terminologies.

What does "stringent fitness" mean exactly? Is it a length of lag phase? Is it the log-enrichment slope of strains carrying relA-spoT knockout?

Specific comments

Line 187: Not clear from the figure. Could be referring to figure 1B, but the antibiotics (blue) line is not at position 526.

Line 250: It is not clear from the text that these are two separate groups of residues (in the figure it is clear). Re-word to make clear these are not the set of residues/mutations that have both increased fitness and stringent enrichment (as in Fig. 2C quadrant 1).

Line 261: What was the effect of hydrophobic hydrophobic mutations? (may have been answered in the supplement, but I did not have access to this).

Line 300: "clustered", I am not so sure by looking at Figure 5C. Seems like red and blue both 'cluster' in the RpoB-RpoC interface.

Figure 5D: why not use the same color scheme for the high mean fitness residues as in Figure 5C?

Line 305-306/Fig 5E: Mutation P560H is not shown relative to BH-777. V558E is not shown in relation to R780.

Line 310: "Other possible interaction-decreasing mutations also improved fitness (Supplementary figure S5A)." Would be nice to have seen the extent of the explanations provided in this Supplementary Figure.

Line 324/Figure 5G: it is unclear from the figure what "covered all residues with high mean growth-associated fitness". The overlap is not clear from the figure, I would suggest using semi-transparent residues to show overlap. The legend for CBR703 is not seen in the figure (presumably because there is overlap?)

Lines 326-329: please label the residue numbers in Figure 5 (as was done well in Fig. 4), especially the ones that are referred to in the main text in these lines.

Lines 358-365: It was very unclear for me to relate it back to Figure 6C. Please make the legend clearer (maybe include the total number of mutants) to make it match the text in these lines.

Line 384-388. It seems like these lines are contradictory/unclear: "The stringent response is 384 absent in archaea and eukaryotes. In both eukaryotes and archaea, the stringent phenotype-determining DNA-proximal motif between residues 531-538 was absent." and immediately after "Therefore, we observed significant conservation of growth-associated residues in proteobacteria and stringent residues in stringent response-associated species across kingdoms."

Line 596: the axis label in the figure is not "stringent enrichment" but "fitness".

Line 602: the axis label in the figure is not "stringent enrichment" but "fitness".

Reviewer #2 (Remarks to the Author):

This is an exciting paper that employed a combination of cutting edge molecular biology techniques. The study showcase how multiple cool techniques such as genome editing, random mutagenesis, and deep mutational scanning, enable us to explore proteins that are traditionally extremely difficult to study. The work studied the effect of RNAP that is central to the biological system, and fairly large and complex molecule, and often observed in experimental evolution to adapt various conditions. The authors cleverly performed the screening in multiple different conditions, which exposed constrains and tradeoffs among mutations. The figures are very well made and easy to understand. The text is also well written and easy to follow their logics. This is a high quality work that teaches us much about the protein and interesting evolutionary dynamics occurring in many other experimental evolution. I believe that this paper deserves a prompt publication in a high profile journal such as Nature Communications.

Reviewer #3 (Remarks to the Author):

In what is in general a very interesting study the authors apply a method they developed (CREPE) that allows them to preform deep mutational scanning (DMS) within essential E. coli genes in their native genomic context, to carry out DMS on a relatively short region of rpoB. rpoB encodes part of the RNA polymerase (RNAP), which was shown to serve as a frequent target for adaptation of E. coli under a

variety of lab-imposed conditions. The specific region of the RNAP examined was selected, because 40% of all known RNAP adaptations were found to occur within that 100 position region. Using CREPE the authors built a library of ~6000 variants in that target region. They then examined the fitness of each member of this library (relative the ancestral genotype) under five conditions: M9 + glucose, M9 + glycerol, M9 + galactose, M9 + glucose + high osmolarity, and M9 + glucose + 0.6% butanol. They found that many of the adaptive mutations show an adaptive effect across the examined conditions (reflected in a strong positive correlation between their fitness across conditions). Reconstruction of many of this variants allowed them to show that they improve growth under the conditions tested. The authors then created another library of variants within a strain of E. coli from which the *relA* and *spoT* genes were deleted. This enabled them to identify mutations within their region of the RNAP that were likely involved in improving the stringent response. As expected a negative correlation was observed between the growth-associated fitness of a mutation and its stringent enrichment, although surprisingly some mutations were found to improve both growth and stringency. The authors then show that each of the two phenotypes (improvement of growth, vs stringency) is associated with a modular cluster of residues that are each involved in a different type of interaction (loss of internal polar interactions for stringency and decreased target-BH interaction for growth). Finally, they show that there need to not be a tradeoff between the two as some variants can do both (the ones that improved both phenotypes), and since combining mutations can lead to both phenotypes together.

My comments:

1) I think this paper suffers from some overselling – From the abstract, introduction and discussion it would seem that the entirety of the RNAP was scanned, when in fact this paper deals with only a specific short region, which was selected because it is the target of adaptation across many conditions. Previous studies have revealed very little overlap in the positions involved in adaptation under different conditions, which may suggest that this region, while very interesting, is not very representative when it comes to the entirety of adaptation that occurs within the RNAP. Similarly, The five conditions tested include some pretty similar conditions, while the RNAP was shown to be involved in adaptation to a much broader range of conditions. As with the region selected for study this may impact the broadness of the claims made (especially with regards to the fact that in this study the fitness effects of the mutations tend to be so well correlated across conditions, which raises interesting questions as to why a this region would be so well conserved in long-term evolution, if so many mutations within it are adaptive under so many conditions). In their discussion the authors suggest that the reason for this is that in nature bacteria need to face fluctuating conditions, where such mutations that are adaptive under one condition may not be under the next. But does this make sense, if these mutations appear to be adaptive under all conditions? This, in my opinion, needs to be better described, caveated and discussed.

2) Clarity to writing – The paper is a bit difficult to follow. When finishing the read the abstract and introduction I found myself a bit lost and could only obtain a good grasp of what was actually done when I finished reading the results. The discussion didn't actually add much discussion, after finishing the results. Mention of growth and maintenance in the abstract and introduction is made to describe the main findings of this paper (I think referring to what later is described as growth vs stringency in the results). But these terms aren't really defined. Growth, is I guess self-explanatory. But maintenance is much less so. This was one of the reasons I think the paper was difficult to follow, but in general I think that given the complex material this paper deals with, more effort should be given to the writing to make things clearer and easier on the reader.

3) Looking at Figure 2 (assuming all scanned variants are included), does it not seem a bit odd that most variants appear to be adaptive? Can this be discussed more? What % of scanned variants were adaptive under each condition tested? (I did not see discussion of this in the paper, aside from a brief discussion of the DFE in section 2.6, which I have to admit I did not quite understand). Is this proportion higher or lower than one would expect? If it is very high (as appears to be the case from section 2.6, looking at figure 2B, and at the DFE given in Figure 5), and if the effects correlate across conditions wouldn't it suggest that somehow this entire region that can change adaptively in most

conditions tested is nevertheless extremely conserved over long term evolution? Why would that be the case? Wouldn't we expect to see some strains of bacteria (or at least of *E. coli*) that are sequenced and that currently have one of these adaptive alleles, if these alleles are indeed adaptive across all or even many conditions? While this is briefly discussed in the discussion section 3.6, I did not find the explanation very satisfying given the apparent results of this study that seem to imply that mutations are adaptive across conditions, can be adaptive to both phenotypes considered, without apparent tradeoffs.

4) In Figure 2, one sees a very strong correlation between the fitness effects of the different variants across conditions. Not only are variants that are adaptive under condition A also adaptive under condition B, there is actually a very nice correlation in the specific fitness effects. This is interesting and requires discussion, as it is not a trivial observation.

Dear Reviewers,

We would really like to thank you for taking the time to make comments on our manuscript. We have addressed your comments below. We formatted the response to the reviewers and editorial recommendations to have the questions raised/recommendations in bold font and our response in regular font. All text changes in the manuscript are in red. The text changes have been italicized in this response to the reviewers for ease of comprehension for the reviewers. The changes made in response to the reviewer's comments have improved the quality of our paper.

Please see our point-by-point review below:

REVIEWER COMMENTS

Reviewer #1 (Remarks to the Author):

The topic of this paper is RNAP mutations that are acquired during evolution and studying their causality. The origin if these studies was the Herring et al paper in Nature Genetics in 2006, where for the first time rpoC mutations from laboratory evolutions were introduced into the starting strain showing their clear causality. Balas Papp et al reviewed this story in a Nature Reviews Genetics paper (I forget the year), and ref 4 did a multiscale study on the mechanisms of causality, except at the structural level where the current study makes a notable contribution. The primary methods used here is to create a library of mutants using targeted mutagenesis and screening the populations of mutants using enrichment cultures. The paper and it results are suitable for Nat Comm, but there are several issues that need to be addressed. It is a very good paper, well written. Few minor grammatical errors.

We really thank the reviewer for the positive feedback on our manuscript and for taking the time to make specific comments. Please find our point-by-point response below:

Larger issues

1. How can cells have both stringent response and growth-promoting mutations simultaneously? My understanding is that these are two opposite forces, and it is impossible to both famine and feast at the same time. If this were possible, it is unclear why E. coli RNAP wouldn't already have said mutations.

This was in fact a surprise for us as well. However, other studies have also reported the ability of bacteria to break tradeoffs during laboratory evolution. We have added citations in our discussion. The famine-feast trade-off mentioned by the reviewer is based on the proposition that limited resources are shared between the two objectives. However, the observation that some mutations/mutation combinations can favor both feast-and famine-associated functions suggests that there is a surplus buffer of resources. Recent studies have also highlighted this surplus of resources. To clarify these points, we have revised our discussion on Page16 lines 17-20 of the revised manuscript:

"It has always been thought that cells cannot improve both growth and stringent regulation due to molecular-level and systems-level constraints. At the systems level, it is proposed that the stringent response controls the allocation of the "limited" resources between growth and maintenance⁷ and a reallocation of the limited resources causes tradeoffs³. However, the ability of cells to both improve growth and have the stringent phenotype shows that resources may not be limited between the two objectives at a systems level. Interestingly other adaptive laboratory evolution experiments have led to evolved E. coli that break growth-survival tradeoffs as well⁵². Several studies have reported the presence of excess resource reserves in exponentially growing cells^{53,54}. "

For the second part of the question, we realize that in our initial version of the manuscript, it came across that only the tradeoffs limit the success of mutations. Since these residues are highly conserved, we wanted to highlight that these residues have functional importance as well. We think that these residues regulate the RNAP function. The presence of this regulatory function is important for

survival in the wild. The loss of the regulatory function is likely lethal. Therefore, the variants with both phenotypes may overcome the tradeoff but will lose two regulatory functions to be counter-selected. We realize that in our previous version, this point was not clear. So, we clarify the point better now in the revised discussion on pages 17-18 of the revised manuscript:

“Despite frequent variation in the laboratory, both the growth and stringent phenotype-determining residues are highly conserved (Figure 7). Residues with high mean stringent enrichment are conserved even across kingdoms i.e., in plant chloroplasts. Each residue is also associated with an interaction and in each case the cells improved fitness by losing the interactions. As opposed to the controlled laboratory evolution, bacteria such as E. coli have naturally evolved to survive in fluctuating environments such as in cycles of feast and famine⁵⁸. Most growth-favoring and stringent mutations were specialists for either growth or stringency and were associated with a tradeoff with respect to the other objective (Figure 2). Such specialist mutations with one objective optimized at the cost of others may be counter-selected in fluctuating environments². While the counterselection of specialists explains the residue conservation of most residues, it does not explain the conservation (specifically within proteobacteria) of residues such as N573 where mutations optimized both objectives.

Residue conservation is also a strong predictor of functional importance³⁵. In addition to performing transcription, the RNAP also regulates global transcription. Regulation requires a change (activation or repression) of gene expression in response to a change in the environment. We propose that the growth-determining and stringent residues and their associated interactions may be important for regulation. The regulatory importance is clear for the stringent residues because the stringent mutations of the RNAP lose the ability to regulate gene expression in response to starvation. Instead, the maintenance functions such as amino acid biosynthesis are constitutively activated in the stringent variants. Similarly, the target-BH interaction likely regulates growth. The small molecule ppGpp is known to regulate transcription elongation for growth-associated ribosome biosynthesis genes by increasing pausing^{59,60}. The pause-resistance of the growth-promoting mutations likely bypass these regulatory pauses to promote growth. Therefore, we propose the described residues and the associated interactions have a regulatory function. The dissonance between residue conservation and laboratory-determined fitness suggests that the associated function is not important in the controlled selection environment and even provides a significant fitness advantage. Similar loss-of-regulation has been previously selected in multiple laboratory evolution experiments⁶¹⁻⁶³. The importance of the regulatory function in the fluctuating environment likely mandates the conservation of the described residues, specifically the ones that overcome tradeoffs (as they are likely associated with loss of two functional interactions)”

We are currently investigating the regulatory role of these mutations. However, the observations are preliminary and out of the scope of this manuscript.

2. A large amount of the work is to demonstrate how variants with substantial phenotypes cluster near or far from each other. However, the figures to demonstrate this are largely qualitative. It would be good to have some more quantitative analysis, like Grantham scores, changes in polarity, or some structural predictions.

We address this question along with question 5 below as both deal with the characterization of mutations using scoring such as the Grantham score. We detail the estimates below.

3. What is the molecular basis of increased stringent fitness? Do stringent genes have different promoter sequences than growth-promoting genes? Is it about interaction with other trans-acting proteins? Why did the increased catalytic rate and pause resistance by growth-promoting mutations not increase expression of stringent genes? The argument that enriched mutations found in a relA and spoT KO'd strain are evidence that stringent-promoting mutations exist in RNAP is a stretch. Removing such a large regulatory element will have very large effects on the transcriptome that will be difficult to detangle. Did you try any other methods such as imposing amino acid starvation on wildtype E. coli?

We thank the reviewer for pointing out that our explanation of the mechanism of stringent response and the justification that the stringent mutations favor maintenance/starvation functions may not have been clear in the text. To clarify further, we changed the section to better explain this selection on page 8 of the revised manuscript on Page 8, lines 1-10:

*“Therefore, we postulated that the target may also contain residues where mutations favor the stringent response and reduce growth. Stringent mutations of the RNAP can be selected by imposing amino acid starvation in the $\Delta relA\Delta spoT$ strain of *E. coli*²⁸. When *E. coli* experiences external stress or starvation, there is an increase in the concentration of the small molecule alarmone ppGpp. ppGpp and the transcription factor DksA bind to the RNAP to induce conformational changes in the RNAP⁹. These conformational changes activate the transcription of genes important for maintenance/starvation response such as amino acid biosynthesis⁹. Therefore, null ppGpp strains, such as $\Delta relA\Delta spoT$ strain of *E. coli*, with both ppGpp synthesis enzymes (*RelA* and *SpoT*) deleted, are auxotroph for amino acids. $\Delta relA\Delta spoT$ strains of *E. coli* cannot grow in minimal media in the absence of amino acids. Stringent mutations of the RNAP mimic the ppGpp-bound state to escape the auxotrophy by constitutively activating the amino acid biosynthesis genes (Supplementary figure S3A)²⁸. Consequently, the stringent mutations of the RNAP favor maintenance/starvation-associated functions.”*

Based on this explanation above, we do not think that the selection in the $\Delta relA\Delta spoT$ strain for maintenance-favoring mutations is a stretch. The absence of ppGpp may alter the global transcriptome. However, as we explain in the text here, ppGpp null strains are auxotroph for amino acids because the RNAP cannot activate amino acid biosynthesis. Therefore, in minimal media without amino acids, the primary selection is for the activation of starvation/stress-associated amino acid biosynthesis. The mutations in the RNAP that mimic the ppGpp-bound state may alter the expression of other ppGpp-regulated genes, but the enrichment score is a direct outcome of survival by activating amino acid biosynthesis.

There could be pleiotropic activation of other stress-associated genes in the stringent mutations as well. We think that such activation may explain a generalist adaptation of stringent mutations to multiple maintenance-associated stresses. We tried to make this point clearer in our text in the previous version, but it was not clear. So, we clarified the same by adding the following changes in our discussion on Page 15 lines 5-17:

*“Growth control is intricately tied to stringent regulation in *E. coli*. Growth-improving mutations had a delayed stringent response (Supplementary figure S2C-D). During the stringent response following an increase in the levels of ppGpp, the binding of ppGpp and dksA to the RNAP activates the expression of starvation/maintenance-associated genes such as the ones involved in amino acid biosynthesis. By imposing a starvation for amino acids in $\Delta relA\Delta spoT$ strain, we also found multiple stringent mutations of the RNAP, that favor amino acid biosynthesis with the trade-off of reduced growth (Figure 2). The stringent mutations mimic the ppGpp-bound state of the RNAP. Therefore, they may also activate the expression of other maintenance functions to again alter global traits. In line with our proposition that RNAP mutations select for altered global traits, it is important to note here that, stringent mutations of the RNAP have also been selected for maintenance-associated adaptation to stresses such as long-term starvation, tolerance, and resistance to antibiotics and chemicals¹²⁻¹⁴.”*

Finally, we already addressed why the increased catalytic rate and pause resistance do not increase the expression of stringent genes. Here we see that the maintenance functions such as amino acid biosynthesis are limited by activation. Therefore, mutations that overcome this activation barrier by mimicking the ppGpp-bound state favor maintenance. While discussing the trade-offs, we propose that catalytic rate and pause resistance may inhibit maintenance-associated functions to explain the observed trade-off that growth-improving mutations have delayed stringent response:

“The tradeoff is likely molecular. It is well known that the decreased open complex intermediate stability of stringent mutations decreases transcription initiation for growth-determining ribosome biosynthesis genes²⁸. Additionally, an increased transcription rate may increase free RNAP availability, which is known to inhibit maintenance-associated genes such as amino acid biosynthesis⁵⁰.”

Additionally, ppGpp may induce pausing specifically to inhibit growth functions such as ribosome biosynthesis. So, pause resistance would allow RNAP to overcome such pauses and increase growth. We discuss the same in the revised manuscript as well on Page 18 lines 12-15:

“The small molecule ppGpp is known to regulate transcription elongation for growth-associated ribosome biosynthesis genes by increasing pausing^{59,60}. The pause-resistance of the growth-promoting mutations likely bypass these regulatory pauses to promote growth.”

At this time, we can only make this hypothesis based on the previous literature. To answer the question precisely, we need to perform more experiments which are unfortunately beyond the scope of this study.

4. A primary concern is that the authors work to demonstrate how variants with substantial phenotypes either cluster near each other or far away from each other. However, the main figures meant to demonstrate this are qualitative (examples: Figure 3B, Figure 4E), and there are better approaches that allow for a more quantitative representation. This more quantitative representation of the distance between variants and their potential clustering would be accomplished through analytics.

We understand that the initial representation of the clusters of residues was qualitative. The initial representation was meant to see if residues, where (several) mutations lead to either of the two phenotypes in question occurred in overlapping residues or separate residues. We see that the non-overlap was demonstrated in Figure 3A. We then just mapped the 2 clusters of residues on the 3D structure of the target. We go on to investigate the possible molecular interactions associated with each phenotype in the subsequent figures 4, and 5. The separation of function was further shown using residue N573 and the double mutants, where both traits could be achieved.

However, we thank the reviewer to bring to our attention that a quantitative analysis of the distance of residues from various interacting partners will strengthen our claims. Therefore, we estimated the distance of each residue from interacting molecules of DNA and RpoC. Then we correlated the growth-associated fitness and the stringent enrichment to each distance. We also performed multiple regression for each fitness and the stringent enrichment to find which variable most explains the change in fitness. We report the findings in Figure 3, supplementary table S4 and in text revisions on Page 9. Also, we understand that the work cluster/module has statistical importance. Therefore, we refrain from using the word cluster/module, until we demonstrate the independence later:

“To uncover the molecular mechanisms underlying these selections, we analyzed the organization of the fitness-conferring mutations within the RNAP structure. The target region interacted with the RNAP RpoC subunit and the DNA (Figure 3B). Univariate and multivariate analysis revealed that the growth-associated fitness was negatively correlated with distance from RpoC but not with distance from DNA (Figure 3C and Supplementary Table S4). In contrast, the stringent enrichment was negatively correlated with distance from DNA but not with distance from RpoC (Figure 3D and Supplementary Table S4). Despite being associated with sequentially and structurally proximal residues, each phenotype may be controlled largely by a specific set of residues, likely associated with two different molecular interactions. ”

Concerning Figure 4E and 5H we *quantitatively* measured the epistasis for all double mutants in the dataset. We then report the residue combinations with significant epistasis (the significance test described in **Supplementary Note S2, Supplementary Figure S4 and S5**). So, we reworded the text to make it clearer in page 11:

“Possible long-range interactions can be inferred by measuring epistasis in double mutants. We estimated the epistasis for all double mutants and evaluated if distant residues had significant positive epistasis between them (Supplementary note S2, Supplementary figure S4D and E). We found significant positive epistasis between the internal and DNA-proximal residues, suggesting the presence of such long-range interactions (Figure 4E, Supplementary note S2, and Supplementary figure S4D and E).”

5. Amino acid substitutions may have different functional/structural consequences depending on the similarities between the original amino acid and the substituting residue. Thus, Fig 4A and Fig 5B may be misleading. That is: Some positions may seem to be not stringent, if there's only substitutions that do not change much (Leucine to isoleucine i.e), although the position is stringent. It would be great if the authors incorporate some kind of measure to estimate the consequence of the substitution (Grantham score maybe?).

This is an excellent point raised by the reviewer. Indeed, it could be possible that due to similarities in substitution, we under-sample the possible changes per residue and end up mischaracterizing some residues. Therefore, to ensure that residues with a wide range of physical-chemical properties were samples, we estimated the recommended Grantham score for all substitutions in both the growth-associated selection and the stringent selection. We added these figures in the supplements and mention this point in the text now.

We added the validation of a range of substitution as a method of validation in the text:

“To make inferences using mean growth-associated fitness and mean stringent enrichment, we also verified that we sampled substitutions with a broad range of physio-chemical properties in the target (Supplementary Note S2).”

Due to the word limit, we describe our validation in Supplementary Note 2:

“Note S2: Validation of mutation diversity

Mutations to residues with similar properties, such as leucine-isoleucine, may have fitness comparable to wild-type. Therefore, if certain positions had only such similar substitutions, their mean growth-associated fitness and mean stringent enrichment would be underestimated due to under-sampling. To verify the diversity of substitutions, we used the Grantham score, which is a measure of amino acid distance based on: composition, polarity, and molecular volume⁵⁴. We mapped the residue-wise distribution of the Grantham score for all substitutions. We scored 5 +/-1 and 8 +/- 2 non-synonymous substitutions per position for growth and stringent enrichment respectively. We observed that substitutions at each position covered a broad distribution of Grantham scores at all positions. Additionally, the distribution of the Grantham score was comparable between stringent versus non-stringent and growth-promoting versus non-growth-promoting residues (**Supplementary Figure S1E and S1F**). ”

We also added supplementary figures **S1E and S1F**.

Smaller issues

Define certain terms - close proximity for example, how many Angstroms is that?

We now have plots to show phenotypes as a function of distance measured as Angstroms.

Clarify certain terminologies.

What does “stringent fitness” mean exactly? Is it a length of lag phase? Is it the log-enrichment slope of strains?

We had explained the stringent fitness in our text. It is the log-enrichment of the mutations in the relA-spoT knockout. However, we mentioned it as “Variant” enrichment. So, we rephrase it to make it clearer on Page 8 lines 17-18:

“We observed significantly more CFUs for the RpoB library in the ΔrelAΔspoT strain compared to a non-edited control upon plating on minimal media without amino acids (Supplementary figure S3B). We measured the variant stringent enrichment (fitness) as a log change in variant frequency before and after plating on M9-Glucose relative to the wild-type control (Supplementary figure S3C). We identified 123 single (and ~418 total) stringent mutations within the target (Supplementary figure

S3B-C and Supplementary note S1). The presence of stringent mutations within the target suggested its role in the stringent response”

Specific comments:

Line 187: Not clear from the figure. Could be referring to Figure 1B, but the antibiotics (blue) line is not at position 526.

We modified the figure to match the legend to the marker.

Line 250: It is not clear from the text that these are two separate groups of residues (in the figure it is clear). Re-word to make clear these are not the set of residues/mutations that have both increased fitness and stringent enrichment (as in Fig. 2C quadrant 1).

We modified the text on Page 10 as per the reviewer’s request. So the entire paragraph is not revised (See question 4 above)

Line 261: What was the effect of hydrophobic hydrophobic mutations? (may have been answered in the supplement, but I did not have access to this).

We added an additional figure comparing hydrophobic and polar, charged, and bulky residues explicitly on page 10 lines 15-16:

*“In this cluster, substitutions in the hydrophobic residues to amino acids with polar, charged, and bulky side chains, which could interfere with DNA interactions, led to the stringent phenotype (with significantly higher stringent enrichment compared to both synonymous mutations and hydrophobic residues (Kruskal Wallis H-test, p-value = 10^{-8} and 10^{-3} respectively), **Supplementary figure S4A and S4B**).”*

We also added a supplementary figure S4B to demonstrate the same:

“A category plot comparing the stringent enrichment scores for mutations to hydrophobic residues to those of residues with polar, charged, and bulky residues.”

Line 300: "clustered", I am not so sure by looking at Figure 5C. Seems like red and blue both 'cluster' in the RpoB-RpoC interface.

The reviewer is correct in pointing this out. Indeed, not all residues at the RpoB-RpoC interface contribute to the fitness, as the fitness has to do with interaction with the bridge helix (BH). Therefore, we rephrased the text as Page 11 lines 25-26:

Several but not all residues with high mean growth-associated fitness clustered around the β - β' (RpoB - RpoC) interaction surface (Figure 5c).

In addition, we added data to show the dependence of growth-associated fitness on the distance from RpoC (Figure 3).

Figure 5D: why not use the same color scheme for the high mean fitness residues as in Figure 5C?

We did not use the same color scheme because we had already used Blue to represent the stringent residues. Therefore, we did so to avoid overlap in subsequent figures.

Line 305-306/Fig 5E: Mutation P560H is not shown relative to BH-777. V558E is not shown in relation to R780.

We thank the reviewer for pointing this out, we fixed the figure 5e.

Line 310: "Other possible interaction-decreasing mutations also improved fitness

(Supplementary figure S5A)." Would be nice to have seen the extent of the explanations provided in this Supplementary Figure.

We had excluded it earlier to manage word counts. We added the extent of explanations in the main text on Page 12 lines 2-6:

“The residue F545 occurred in close proximity to a lysine, K789, residue. Aromatic rings close to positively charged residues can form cation-pi interactions. Mutations that may disrupt the possible interaction also improved growth-associated fitness (Supplementary figure S5A).”

Line 324/Figure 5G: it is unclear from the figure what "covered all residues with high mean growth-associated fitness". The overlap is not clear from the figure, I would suggest using semi-transparent residues to show overlap. The legend for CBR703 is not seen in the figure (presumably because there is overlap?)

We agree with the reviewer. With the way the figure was represented, it was not clear as to which residues were growth-associated and which ones were resistant to CBR703. If we compare the old figures 5D and 5G, there were growth-associated residues that were not resistant to CBR703. Therefore, to clarify, we represent all residues as spheres: The ones resistant to CBR703 and also growth-associated with the color pink, only growth-associated ones with the color green, and ones only resistant to CBR703 as white. Based on the reviewer's recommendation, we also changed the transparency of the spheres and clarified the legend better. We change the text as well on Page 12 lines 17-18:

“CB5R703 resistance, estimated as the log fold- enrichment for growth in CBR703, correlated strongly with the growth-associated fitness (Figure 5f, Keio: $\rho = 0.5$ and p -value $< 10^{-16}$); and most resistance-conferring residues had high mean growth-associated fitness (Figure 5g).”

It is important to note here that for the non-overlapping residues, we still see a correlation between the growth-associated fitness and CBR703 resistance (Figure 5e).

Lines 326-329: please label the residue numbers in Figure 5 (as was done well in Fig. 4), especially the ones that are referred to in the main text in these lines.

We added the labels in Figure 5.

Lines 358-365: It was very unclear for me to relate it back to Figure 6C. Please make the legend clearer (maybe include the total number of mutants) to make it match the text in these lines.

We fixed the text to better correlate to the plot on page 13: lines 19-26:

“A majority, 70.3% (64/91), of such combinations had both the growth-improving and the stringent phenotype (Blue, Figure 6c). Of 138 double mutants with both high growth and the stringent phenotype, 46% (64/138) variants were a combination of stringent and growth-improving mutations (Blue, Figure 6c). Additionally, 76.1% (105/138) variants occurred in positions where one position had high mean growth-associated fitness and the other had high mean stringent enrichment (Pink and Blue, Figure 6c).”

We fixed the figure legend as well:

“Correlation of growth-associated fitness and stringent enrichment all double mutations (grey), and ones with combined synonymous mutations (green), combined growth-improving and stringent mutations (64, blue), and combined mutations on positions important for growth and stringent phenotypes (41, pink. 105 total combined with blue).”

Line 384-388. It seems like these lines are contradictory/unclear: "The stringent response is 384 absent in archaea and eukaryotes. In both eukaryotes and archaea, the stringent phenotype-determining DNA-proximal motif between residues 531-538 was absent." and immediately after

"Therefore, we observed significant conservation of growth-associated residues in proteobacteria and stringent residues in stringent response-associated species across kingdoms."

We fixed the line in Page 13 lines 27-28:

"Therefore, we observed significant conservation of growth-associated residues in proteobacteria and stringent residues in stringent response-associated species across kingdoms in bacteria and plant chloroplasts"

Line 596: the axis label in the figure is not "stringent enrichment" but "fitness".

We fixed the y-axis in the figure

Line 602: the axis label in the figure is not "stringent enrichment" but "fitness".

We fixed the y-axis in the figure

Reviewer #2 (Remarks to the Author):

This is an exciting paper that employed a combination of cutting-edge molecular biology techniques. The study showcase how multiple cool techniques such as genome editing, random mutagenesis, and deep mutational scanning, enable us to explore proteins that are traditionally extremely difficult to study. The work studied the effect of RNAP that is central to the biological system, and fairly large and complex molecule, and often observed in experimental evolution to adapt various conditions. The authors cleverly performed the screening in multiple different conditions, which exposed constrains and tradeoffs among mutations. The figures are very well made and easy to understand. The text is also well written and easy to follow their logics. This is a high-quality work that teaches us much about the protein and interesting evolutionary dynamics occurring in many other experimental evolution. I believe that this paper deserves a prompt publication in a high-profile journal such as Nature Communications.

We would really like to thank Reviewer 2 for the generous feedback and appreciate that the message of the paper was clear. We are happy to see that the reviewer understood the message of the paper well.

Reviewer #3 (Remarks to the Author):

In what is in general a very interesting study the authors apply a method they developed (CREPE) that allows them to preform deep mutational scanning (DMS) within essential E. coli genes in their native genomic context, to carry out DMS on a relatively short region of rpoB. rpoB encodes part of the RNAP (RNAP), which was shown to serve as a frequent target for adaptation of E. coli under a variety of lab-imposed conditions. The specific region of the RNAP examined was selected, because 40% of all known RNAP adaptations were found to occur within that 100 position region. Using CREPE the authors built a library of ~6000 variants in that target region. They then examined the fitness of each member of this library (relative the ancestral genotype) under five conditions: M9 + glucose, M9 + glycerol, M9 + galactose, M9 + glucose + high osmolarity, and M9 + glucose + 0.6% butanol. They found that many of the adaptive mutations show an adaptive effect across the examined conditions (reflected in a strong positive correlation between their fitness across conditions). Reconstruction of many of this variants allowed them to show that they improve growth under the conditions tested. The authors then created another library of variants within a strain of E. coli from which the relA and spoT genes were deleted. This enabled them to identify mutations within their region of the RNAP that were likely involved in improving the stringent response. As expected a negative correlation was observed between the growth-associated fitness of a mutation and its stringent enrichment, although surprisingly some mutations were found to improve both growth and

stringency. The authors then show that each of the two phenotypes (improvement of growth, vs stringency) is associated with a modular cluster of residues that are each involved in a different type of interaction (loss of internal polar interactions for stringency and decreased target-BH interaction for growth). Finally, they show that there need to not be a tradeoff between the two as some variants can do both (the ones that improved both phenotypes), and since combining mutations can lead to both phenotypes together.

We really thank the reviewer for their feedback. We address the concerns point by point below:

My comments:

1) I think this paper suffers from some overselling – From the abstract, introduction and discussion it would seem that the entirety of the RNAP was scanned, when in fact this paper deals with only a specific short region, which was selected because it is the target of adaptation across many conditions. Previous studies have revealed very little overlap in the positions involved in adaptation under different conditions, which may suggest that this region, while very interesting, is not very representative when it comes to the entirety of adaptation that occurs within the RNAP. Similarly, the five conditions tested include some pretty similar conditions, while the RNAP was shown to be involved in adaptation to a much broader range of conditions. As with the region selected for study this may impact the broadness of the claims made (especially with regards to the fact that in this study the fitness effects of the mutations tend to be so well correlated across conditions, which raises interesting questions as to why a this region would be so well conserved in long-term evolution, if so many mutations within it are adaptive under so many conditions). In their discussion the authors suggest that the reason for this is that in nature bacteria need to face fluctuating conditions, where such mutations that are adaptive under one condition may not be under the next. But does this make sense, if these mutations appear to be adaptive under all conditions? This, in my opinion, needs to be better described, caveated and discussed.

We thank the reviewer for the remarks. We did not intend to oversell our manuscript. We understand that the comment implies that it may come across that we scanned the entirety of the RNAP in certain portions of the paper. Therefore, we rephrase certain regions of the paper to make the point explicit:

Abstract:

We changed “*We, therefore, measured the fitness of thousands of *rpoB* variants under multiple conditions and genetic backgrounds*” to “*We, therefore, measured the fitness of thousands of mutations within a region of *rpoB* under multiple conditions and genetic backgrounds*”

Introduction:

At the end of the introduction, we already explicitly mention that scanning was done over a region of the RNAP (Page 5, lines 15-18):

*“By analyzing adaptive mutation databases, we identified a region within the *RpoB* subunit of the RNAP where mutations improved fitness in multiple environments. We used CREPE to generate a rich library of 6000 variants with mutations targeting this region and measured the fitness of the variants in multiple environments (Figure 1A).”*

We could not explicitly describe this region earlier in the introduction because the discovery of the region was a result of the analysis of ALE databases. So, we deemed it fit to discuss it in results section 1.

Previous studies, cited in our manuscript, have made claims on adaptive mechanisms of the RNAP using only a handful of mutations (often even 1-2 mutations). Such mutations are often non-overlapping as the reviewer mentions, to make broader arguments on pleiotropy and trade-offs. Therefore, the few mutations provide only a limited view of the adaptive landscape of the RNAP. We wanted a deeper insight into RNAP-mediated adaptation and pleiotropy using deep mutational scans. However, with methods like CREPE, mutational scans can cover only scan a small region (100-120 amino acids) of the

genome. In terms of technologies for throughput genomic mutational scans, this is the best we can do and is already a significant improvement on previously published work. Since, we found that the mutations in the target region enabled versatile adaptation, and were reported to be pleiotropic, we targeted the region to maximize our chances of procuring meaningful data and try and understand possible adaptation and pleiotropy mechanisms.

We extend our findings to other parts of the RNAP only when 1) we have experimental evidence or 2) observations in other parts of the polymerase align with our findings.

For instance, we found that an increase in translation rate and pause resistance may be associated with improved growth. To broaden the scope of this observation, we reconstructed mutations known to increase translation rate and pause resistance and observe an increased growth. Consequently, we support the observation with more data. This observation is described in the line (Page 12: lines 30-33):

“To further test the hypothesis, we constructed two mutations in the RpoC subunit, I774T and I755V, known to increase catalysis rate and pause resistance⁶⁶ (Supplementary figure S6B). We observed that both mutations significantly improved growth compared to the wild-type variant (Supplementary figure S6B).”

We further reinforce the argument with evidence from the literature in the discussion:

“Mutations in other regions of the RNAP known to increase elongation and pause resistance also improved growth (Supplementary figure S6). Other independent in vitro biochemical studies have shown an increased elongation rate for growth-improving mutations^{66,71}.”

In the discussion, we did make some broader claims surrounding the applications of such mutations. We have now excluded the following paragraphs from our revised discussion:

“Mutations in RNAP can also improve tolerance to industrial environments for applications in biotechnology^{37,79-81}. However, improved growth in the industrial environment does not always improve product yields⁸². Improved growth and improved production may be associated with different global traits. For instance, amino acid pathways make precursors for some industrial compounds, and improved growth affects the expression of amino acid biosynthesis genes (Figure 2 and Supplementary figure S2). We could test how combined growth-stringent mutations at positions, such as 573, fare in production.

The target region is also highly conserved across bacteria. Therefore, it is worth testing if the modules allow predictable control of growth or maintenance in other bacterial pathogens and industrial hosts as well.”

P.S.: Finally, The part of the reviewer’s comment asking to explain the dissonance between conservation and cross-environment adaptation is addressed with question 3 at the end of the document due to their similarity.

2) Clarity to writing – The paper is a bit difficult to follow. When finishing the read the abstract and introduction I found myself a bit lost and could only obtain a good grasp of what was actually done when I finished reading the results. The discussion didn’t actually add much discussion, after finishing the results. Mention of growth and maintenance in the abstract and introduction is made to describe the main findings of this paper (I think referring to what later is described as growth vs stringency in the results). But these terms aren’t really defined. Growth, is I guess self-explanatory. But maintenance is much less so. This was one of the reasons I think the paper was difficult to follow, but in general I think that given the complex material this paper deals with, more effort should be given to the writing to make things clearer and easier on the reader.

We intended to keep the introduction quite broad because when we started the project little understanding of the possible molecular mechanisms of adaptation or pleiotropy associated with RNAP mutations. Due to the lack of such studies, we could not find ideal references to open our later

claims as we thought it would be confusing for the readers as to what is known and what is being demonstrated.

We do agree with reviewers 3 that the definition of the Maintenance phenotype was not clear. So, we clearly explain the selection in our results (Page 8 lines 1-10) and discussions:

Results:

*“Therefore, we postulated that the target may also contain residues where mutations favor the stringent response and reduce growth. Stringent mutations of the RNAP can be selected by imposing amino acid starvation in the $\Delta relA\Delta spoT$ strain of *E. coli*²⁸. When *E. coli* experiences external stress or starvation, there is an increase in the concentration of the small molecule alarmone ppGpp. ppGpp and the transcription factor DksA bind to the RNAP to induce conformational changes in the RNAP⁹. These conformational changes activate the transcription of genes important for maintenance/starvation response such as amino acid biosynthesis⁹. Therefore, null ppGpp strains, such as $\Delta relA\Delta spoT$ strain of *E. coli*, with both ppGpp synthesis enzymes (*RelA* and *SpoT*) deleted, are auxotroph for amino acids. $\Delta relA\Delta spoT$ strains of *E. coli* cannot grow in minimal media in the absence of amino acids. Stringent mutations of the RNAP mimic the ppGpp-bound state to escape the auxotrophy by constitutively activating the amino acid biosynthesis genes (Supplementary figure S3A)²⁸. Consequently, the stringent mutations of the RNAP favor maintenance/starvation-associated functions.”*

Discussion:

*“Growth control is intricately tied to stringent regulation in *E. coli*. Growth-improving mutations had a delayed stringent response (Supplementary figure S2C-D). During the stringent response following an increase in the levels of ppGpp, the binding of ppGpp and dksA to the RNAP activates the expression of starvation/maintenance-associated genes such as the ones involved in amino acid biosynthesis. By imposing starvation for amino acids in $\Delta relA\Delta spoT$ strain, we also found multiple stringent mutations of the RNAP, that favor amino acid biosynthesis with the trade-off of reduced growth (Figure 2). The stringent mutations mimic the ppGpp-bound state of the RNAP. Therefore, they may also activate the expression of other maintenance functions to again alter global traits. In line with our proposition that RNAP mutations select for altered global traits, it is important to note here that, stringent mutations of the RNAP have also been selected for maintenance-associated adaptation to stresses such as long-term starvation, tolerance, and resistance to antibiotics and chemicals¹²⁻¹⁴.”*

In addition, we have added more explanations and references in the discussion based on the remarks from all reviewers. We hope these changes have helped improve the clarity of the paper for the reviewer.

PS: Due to its use in question 3, we answer the 4th question before the 3rd one.

4) In Figure 2, one sees a very strong correlation between the fitness effects of the different variants across conditions. Not only are variants that are adaptive under condition A also adaptive under condition B, there is actually a very nice correlation in the specific fitness effects. This is interesting and requires discussion, as it is not a trivial observation.

One of the main demonstrations of the paper was that mutations within this region of the RNAP promote a generalist adaptation by improving “growth” in the M9 media. We now explain it more clearly in the results (Page 7, lines 16-20):

“The variant fitness was strongly positively correlated across environments (Figure 2B). Therefore, the beneficial target mutations were generalist because they improved fitness in multiple environments. The beneficial mutations likely impacted a common global trait, as opposed to condition-specific traits. We individually reconstructed several beneficial mutations and found that each beneficial mutant had an increased growth rate in the minimal media used for adaptation (Supplementary figure S2A and S2B). Therefore, we posited that the target region was involved in RNAP-mediated growth control.”

Thanks to the review, we realized that this point may not have been clear in the discussion section. Therefore, we elaborate on this observation based on the reviewer’s recommendation. Such cross-stress

adaptation has been observed previously during laboratory adaptation studies as well albeit with only 2-3 mutations. So, we add those citations as well (Page 15, lines 18-27):

“Mutations in the RNAP provide access to complex phenotypes and are frequently selected in adaptive laboratory evolution to diverse stresses. RNAP mutations are highly pleiotropic^{3,26}, which made it difficult to identify the traits under selection during adaptive evolution. The current dominant hypothesis is that RNAP may rewire transcription for condition-specific adaptation. However, within the scanned area, the fitness of the beneficial RNAP mutations strongly correlated between five diverse environments (Figure 2), where the only commonality was the base M9-media. Accordingly, reconstructed mutations had a higher growth rate in M9 minimal media compared to wild-type (Supplementary figure S2). Therefore, by measuring the fitness of over 6000 target mutations in multiple environments, we demonstrate that during laboratory evolution selection may favor RNAP mutations that impact global traits such as growth as opposed to previously proposed condition-specific adaptation (Figure 2). Similar cross-environment adaptation has been reported previously for other RNAP mutations^{3,10,26}. Mutations within the region found during ALE in other conditions also had high growth-associated fitness in our analyses (Supplementary Table S1). This suggests that the selection for improved growth extends beyond our tested conditions.

Growth control is intricately tied to stringent regulation in E. coli. Growth-improving mutations had a delayed stringent response (Supplementary figure S2C-D). During the stringent response following an increase in the levels of ppGpp, the binding of ppGpp and dksA to the RNAP activates the expression of starvation/maintenance-associated genes such as the ones involved in amino acid biosynthesis. By imposing starvation for amino acids in $\Delta relA\Delta spoT$ strain, we also found multiple stringent mutations of the RNAP, that favor amino acid biosynthesis with the trade-off of reduced growth (Figure 2). The stringent mutations mimic the ppGpp-bound state of the RNAP. Therefore, they may also activate the expression of other maintenance functions to again alter global traits. In line with our proposition that RNAP mutations select for altered global traits, it is important to note here that, stringent mutations of the RNAP have also been selected for maintenance-associated adaptation to stresses such as long-term starvation, tolerance, and resistance to antibiotics and chemicals¹²⁻¹⁴.”

3) Looking at Figure 2 (assuming all scanned variants are included), does it not seem a bit odd that most variants appear to be adaptive? Can this be discussed more? What % of scanned variants were adaptive under each condition tested? (I did not see discussion of this in the paper, aside from a brief discussion of the DFE in section 2.6, which I have to admit I did not quite understand). Is this proportion higher or lower than one would expect? If it is very high (as appears to be the case from section 2.6, looking at figure 2B, and at the DFE given in Figure 5), and if the effects correlate across conditions wouldn't it suggest that somehow this entire region that can change adaptively in most conditions tested is nevertheless extremely conserved over long-term evolution? Why would that be the case? Wouldn't we expect to see some strains of bacteria (or at least of *E. coli*) that are sequenced and that currently have one of these adaptive alleles, if these alleles are indeed adaptive across all or even many conditions? While this is briefly discussed in the discussion section 3.6, I did not find the explanation very satisfying given the apparent results of this study that seem to imply that mutations are adaptive across conditions, can be adaptive to both phenotypes considered, without apparent tradeoffs.

It is indeed odd that so many mutations were beneficial in multiple tested conditions. Historically, beneficial mutations are expected to be extremely rare because benefit is associated with a gain of function and mutations are more likely to disrupt function than improve it. However, in our case, the cellular-level gain of function was associated with a protein-level decrease/loss of interaction, which we talk about for each phenotype in Figures 4 and 5. For instance, as we demonstrate in Figure 5, a decreased interaction between the target and the BH increased growth-associated fitness. Since multiple mutations can decrease an interaction, we observe the prominent number of beneficial mutations in the DFE. We understand that the point was not clear in the previous draft of the paper. So, we make this argument more explicit by adding a paragraph in our discussion section (Pages 16-17: lines 28-32, and lines 1-5 respectively):

“We found that the global traits of growth and maintenance were controlled by two modular residue clusters, each associated with the loss of a specific molecular interaction (Figures 4 and 5). The coupling of the improved fitness with a loss of molecular interactions led us to an atypical distribution of fitness effects, with a prominent mode of beneficial mutations (Figure 5a). The higher fraction of beneficial mutations results occurs as molecular interactions can be decreased in multiple ways (as mutations are more likely to decrease than increase function), as opposed to the rare beneficial mutations observed in previous protein DFE⁴⁰ that resort from rare gain-of-function.

What are the involved molecular interactions? The stringent phenotype is known to be determined by the stability of an open complex intermediate during initiation, controlled by target-DNA interactions^{28,39}. Accordingly, modifications in DNA-proximal residues (within 10 Angstroms of the target) and internal pairwise interactions (likely associated with open complex stability) were stringent (Figure 4). The growth-associated fitness was linked to a decreased interaction between the target-BH (Figure 5). The decreased target-BH interaction increases the transcription elongation speed and pause resistance during elongation⁴². Mutations in other regions of the RNAP known to increase elongation and pause resistance also improved growth (Supplementary figure S6). Similarly, growth-improving mutations in other regions of the RNAP have also been found to have an increased elongation rate and pause resistance^{42,47-49}. Therefore, an increase in catalysis and pause resistance increases the growth rate. An increase in elongation speed may increase the concentration of free RNAP, which is known to favor growth⁵⁰. Additionally, transcriptional pauses are known to limit elongation for growth-associated genes⁵¹. We cannot completely exclude the possibility that the target-DNA and target-BH interaction may alter multiple other steps during transcription⁴⁹. However, the functions controlled by the target-DNA/internal and target-BH interactions have to be modular because upon combining the loss of internal interaction and reduced target-BH interaction we obtain variants with both an improved growth and the stringent phenotype (Figure 6). If both interactions affected overlapping functions, their independent optimization would not be possible.”

It is indeed interesting that the residues are highly naturally conserved. The conservation of the residues implies that the residues and the associated interactions are important for the RNAP function. What we meant to imply was that these regions have a regulatory function. Loss of the regulation leads to very specific optimized phenotypes with tradeoffs that are beneficial in a controlled laboratory environment but are naturally very important. We did not want to just imply that the residues are conserved due to tradeoffs. We do agree with the reviewer one and three that just the tradeoff argument does not explain variants with multiple phenotypes. To improve the clarity, we have revised the section and added more references to strengthen our arguments:

“Despite frequent variation in the laboratory, both the growth and stringent phenotype-determining residues are highly conserved (Figure 7). Residues with high mean stringent enrichment are conserved even across kingdoms i.e., in plant chloroplasts. Each residue is also associated with an interaction and in each case the cells improved fitness by losing the interactions. As opposed to the controlled laboratory evolution, bacteria such as E. coli have naturally evolved to survive in fluctuating environments such as in cycles of feast and famine⁵⁸. Most growth-favoring and stringent mutations were specialists for either growth or stringency and were associated with a tradeoff with respect to the other objective (Figure 2). Such specialist mutations with one objective optimized at the cost of others may be counter-selected in fluctuating environments². While the counterselection of specialists explains the residue conservation of most residues, it does not explain the conservation (specifically within proteobacteria) of residues such as N573 where mutations optimized both objectives.

Residue conservation is also a strong predictor of functional importance³⁵. In addition to performing transcription, the RNAP also regulates global transcription. Regulation requires a change (activation or repression) of gene expression in response to a change in the environment. We propose that the growth-determining and stringent residues and their associated interactions may be important for regulation. The regulatory importance is clear for the stringent residues because the stringent mutations of the RNAP lose the ability to regulate gene expression in response to starvation. Instead, the maintenance functions such as amino acid biosynthesis are constitutively activated in the stringent variants. Similarly, the target-BH interaction likely regulates growth. The small molecule ppGpp is known to regulate transcription elongation for growth-associated ribosome biosynthesis genes by

increasing pausing^{59,60}. The pause-resistance of the growth-promoting mutations likely bypass these regulatory pauses to promote growth. Therefore, we propose the described residues and the associated interactions have a regulatory function. The dissonance between residue conservation and laboratory-determined fitness suggests that the associated-function is not important in the controlled selection environment and even provide a significant fitness advantage. Similar loss-of-regulation has been previously selected in multiple laboratory evolution experiments⁶¹⁻⁶³. The importance of the regulatory function in the fluctuating environment likely mandates the conservation of the described residues, specifically the ones that overcome tradeoffs (as they are likely associated with loss of two functional interactions).”

REVIEWERS' COMMENTS

Reviewer #1 (Remarks to the Author):

All of the issues previously cited were resolved outside of their assumption that some of the mutations they found both increase stress-adaptation and growth. Their response was that there is an excess pool of resources so cells can optimize both at the cost of this pool and that the mutations that favor both stress-adaptation and growth are not naturally found in wildtype because they have a regulatory function that is important in natural fluctuating environments.

While these points hold some weight, I do not think they are substantial enough to support what they are claiming. The *relA/spoT* KO will likely introduce a multitude of many other changes which further casts some doubt on the results. There is little evidence, even computational, that their mutation affects regulatory mechanisms as they suggest.

I do think the results are interesting and the mutations deserve to be mentioned, but I think another study targeted largely at just these select mutations would be necessary to make the claims they are. If they reduced their certainty surrounding these claims, I would support accepted this paper.

Reviewer #2 (Remarks to the Author):

The authors responded well to the reviewers' comments and the manuscript is improved. I believe that the manuscript is ready to publish in Nature Communications.

Reviewer #3 (Remarks to the Author):

As with the original manuscript, I found the revised manuscript to be very interesting and ultimately worth of publication.

While the authors addressed some of my concerns, there are some they did not address sufficiently in my opinion:

1) I do still feel that the authors do not sufficiently discuss how what they show for the specific region of the RNAP they scan may differ from what may occur in the remainder of the protein complex. The region focused on is one where adaptations seem to overlap many conditions, something that in general is not true for the entire RNAP, for which a general trend was observed by which very little overlap is seen in the identity of specific residues involved in adaptation to various selective pressures. I think this should be implicitly discussed. The authors did improve things by writing in the beginning of the discussion that the dominant hypothesis is that adaptations to the RNAP rewire transcription for condition specific adaptation, however, within the scanned area effects imply a more general function. However, I still think the discussion could benefit from a paragraph in which the possible differences between this specific region of the RNAP and the remainder of the complex are discussed. This paragraph should, in my opinion, include an implicit statement that it is quite possible that adaptations to other regions of the complex, in which adaptations appear to be much more condition specific, could still act via condition specific adaptive alterations to transcription, and that more will be needed to be done to understand whether or not this is the case.

2) I may be missing something, but I don't think the authors actually addressed the 4th comment of my previous review of their manuscript ("In Figure 2, one sees a very strong correlation between the fitness effects of the different variants across conditions. Not only are variants that are adaptive under

condition A also adaptive under condition B, there is actually a very nice correlation in the specific fitness effects. This is interesting and requires discussion, as it is not a trivial observation.”) While they did respond to this comment, I don’t think their response actually answered my question, of why the fitness effects of various variants correlate so well across conditions. I understand their explanation of why it would be that they would correspond in their general effect (e.g. be, or not be adaptive under the various conditions tested), But still don’t understand why we should expect to find such a strong correlation in the extent of the effects of each mutation across conditions).

3) With regards to my third comment, the authors did not address my request for a better description of the distribution of fitness effects of the different mutations tested. What proportion of these mutations were adaptive, what proportion deleterious? How does this compare to other experiments? While the authors do add a discussion of some of this, the actual numbers are still not available, as far as I could find.

Finally, I found what I think is a minor error I did not notice in my original reading of the manuscript, in the Introduction, starting at line 79: “The fitness costs associated with rifampicin resistance-conferring clinical RNAP mutations have led to the emergence of extreme drug resistance and multidrug resistance strains” - This sentence doesn’t make sense to me. Did the fitness costs lead to drug resistance, or did the mutations themselves?

Dear Reviewers,

We would really like to thank you for taking the time to make comments on our manuscript. We have addressed your comments below. We formatted the response to the reviewers and editorial recommendations to have the questions raised/recommendations in bold font and our response in regular font. All text changes in the manuscript are in red. The text changes have been italicized in this response to the reviewers for ease of comprehension for the reviewers. The changes made in response to the reviewer's comments have improved the quality of our paper.

Please see our point-by-point review below:

REVIEWER COMMENTS

Reviewer #1 (Remarks to the Author):

All of the issues previously cited were resolved outside of their assumption that some of the mutations they found both increase stress-adaptation and growth. Their response was that there is an excess pool of resources so cells can optimize both at the cost of this pool and that the mutations that favor both stress-adaptation and growth are not naturally found in wildtype because they have a regulatory function that is important in natural fluctuating environments. While these points hold some weight, I do not think they are substantial enough to support what they are claiming. The *relA/spoT* KO will likely introduce a multitude of many other changes which further casts some doubt on the results. There is little evidence, even computational, that their mutation affects regulatory mechanisms as they suggest. **I do think the results are interesting and the mutations deserve to be mentioned, but I think another study targeted largely at just these select mutations would be necessary to make the claims they are. If they reduced their certainty surrounding these claims, I would support accepted this paper.**

We thank the reviewer for their comment. We do agree that in order to claim a functional role, we need to conduct more experiments. However, as they point out, these would be a part of another extensive study. We realized based on the reviewer's comments that our previous discussion came across as claims. In order to present our idea as a hypothesis, we have rephrased the paragraph to reduce the certainty around the claim that the mutations have a regulatory role. We also state explicitly that more work is required to validate the function of the site.:

“The growth-determining and stringent residues and their associated interactions likely have functional importance, as residue conservation is also a strong predictor of functional importance³⁵. Since, each phenotype is associated with a loss-of-function, it is likely that the functions associated with each residue cluster is important for survival in the wild. The residues may be important for regulation. In addition to performing transcription, the RNAP also regulates global transcription. Regulation requires a change (activation or repression) of gene expression in response to a change in the environment. **However,** the maintenance functions such as amino acid biosynthesis are constitutively activated in the stringent variants. **Therefore, the stringent mutations lose the ability to regulate gene expression in response to starvation, suggesting the regulatory role of the stringent residues.** Similarly, the target-BH interaction may regulate growth. The small molecule ppGpp is known to regulate transcription elongation for growth-associated ribosome biosynthesis genes by increasing pausing^{59,60}. The pause-resistance of the growth-promoting mutations likely bypass these regulatory pauses to promote growth. Therefore, we propose the described residues and the associated interactions **may** have a regulatory function. The dissonance between residue conservation and laboratory-determined fitness suggests that the associated function is not important in the controlled selection environment and even provides a significant fitness advantage. Similar loss-of-regulation has been previously selected in multiple laboratory evolution experiments⁶¹⁻⁶³. **Further, biochemical and computational characterization of the mutations is required to understand the**

functional/regulatory role of the residues. We also need to characterize the mutations in a common background because the $\Delta relA\Delta spoT$ background may introduce unpredictable global changes.”

Reviewer #2 (Remarks to the Author):

The authors responded well to the reviewers' comments and the manuscript is improved. I believe that the manuscript is ready to publish in Nature Communications.

We thank the reviewer for taking the time to review the revised manuscript and the response to the reviewers and their positive feedback.

Reviewer #3 (Remarks to the Author):

As with the original manuscript, I found the revised manuscript to be very interesting and ultimately worth of publication. While the authors addressed some of my concerns, there are some they did not address sufficiently in my opinion:

1) I do still feel that the authors do not sufficiently discuss how what they show for the specific region of the RNAP they scan may differ from what may occur in the remainder of the protein complex. The region focused on is one where adaptations seem to overlap many conditions, something that in general is not true for the entire RNAP, for which a general trend was observed by which very little overlap is seen in the identity of specific residues involved in adaptation to various selective pressures. I think this should be implicitly discussed. The authors did improve things by writing in the beginning of the discussion that the dominant hypothesis is that adaptations to the RNAP rewire transcription for condition specific adaptation, however, within the scanned area effects imply a more general function. However, I still think the discussion could benefit from a paragraph in which the possible differences between this specific region of the RNAP and the remainder of the complex are discussed. This paragraph should, in my opinion, include an implicit statement that it is quite possible that adaptations to other regions of the complex, in which adaptations appear to be much more condition specific, could still act via condition specific adaptive alterations to transcription, and that more will be needed to be done to understand whether or not this is the case.

We thank the reviewer for the comment. We do agree that in other regions of the RNA polymerase, there could be more condition-specific adaptation. We did not intend to state that growth and maintenance control are the only axes of adaptation. To clarify the comment, we added the following paragraph:

“Within the scanned region of the RNA polymerase, we identified modules for global growth and maintenance control, that may be associated with transcription elongation and initiation respectively. However, the RNA polymerase complex also undergoes environment-specific conformational and functional changes to alter transcription. While mutations in other regions of the RNA polymerase affecting elongation speed/pausing and open initiation complex stability may affect growth and maintenance, a condition-specific adaptation via mutations in other regions of the RNA polymerase is also possible. Deep mutational scanning of the RNA polymerase in multiple environments and genetic backgrounds using technologies such as CREPE provides a strong platform to discover such modules associated with the different

environments (Figure 1a). In addition, e discovery of such modules opens lucrative avenues for predictable cellular control; with significant applications in drug design, to prevent propagation of antibiotic resistance, and enable strain engineering in biotechnology and synthetic biology.”

2) I may be missing something, but I don't think the authors actually addressed the 4th comment of my pervious review of their manuscript (“In Figure 2, one sees a very strong correlation between the fitness effects of the different variants across conditions. Not only are variants that are adaptive under condition A also adaptive under condition B, there is actually a very nice correlation in the specific fitness effects. This is interesting and requires discussion, as it is not a trivial observation.”) While they did respond to this comment, I don't think their response actually answered my question, of why the fitness effects of various variants correlate so well across conditions. I understand their explanation of why it would be that they would correspond in their general effect (e.g. be, or not be adaptive under the various conditions tested), But still don't understand why we should expect to find such a strong correlation in the extent of the effects of each mutation across conditions).

We thank the reviewer for the comment. The fitness in each case is not the absolute fitness but relative fitness. As we see in supplementary figure S2, the absolute growth rate for variants between conditions are not comparable. As described in the methods, the fitness was measured relative to the wild-type reference. When we measure the relative fitness compared to the wildtype again for the actual growth measurements for individually reconstructed mutations, we again observe that this relative fitness is correlated with a spearman correlation coefficient of 0.56 (condition is NaCl or Galactose):

For two mutations the fitness of Galactose is higher compared to Glucose and NaCl. We observe that this trend corroborated Figure 2, as the fitness for Galactose compared to glucose falls above the $x=y$ line.

We measured spearman rank correlation for the variants between environments. A strong rank correlation shows that the variant with most benefit in condition A relative to wildtype also has the most benefit in condition b relative to wildtype. If the underlying mechanism of adaptation is the same, as mentioned in the text, we would expect that the the degree of benefit conferred by same mutation in different environments to be the comparable as well.

We rephrased our paragraph in discussion to stress on this point further:

“However, within the scanned area, the fitness of the beneficial RNAP mutations strongly correlated between five diverse environments (Figure 2). Not only were the same mutations adaptive in multiple conditions, the degree of adaptation for a beneficial variant between conditions was also comparable (Figure 2). Therefore, a common underlying mechanism improved fitness across conditions. The only commonality was the between the

conditions was base M9-media **and** reconstructed mutations had a higher growth rate in M9 minimal media compared to wild-type (**Supplementary Fig. 2**).”

3) With regards to my third comment, the authors did not address my request for a better description of the distribution of fitness effects of the different mutations tested. What proportion of these mutations were adaptive, what proportion deleterious? How does this compare to other experiments? While the authors do add a discussion of some of this, the actual numbers are still not available, as far as I could find.

We apologize to the reviewer for the oversight. We have now explicitly added the fraction of mutations occurring in each class and report how they compare with other DFEs observed previously:

*“To understand the growth phenotype, we first looked at the Distribution of Fitness Effects or the DFE. The DFE is a histogram of the frequency of the measured fitness effect size. A DFE almost universally shows an exponentially diminishing positive tail because beneficial mutations are rare **and account for ~0.01-2% of mutations**^{1,40}. On the contrary, the DFE for the growth-associated fitness had a prominent beneficial mutation mode (**Figure 5a**). **Amongst all unique single mutations scored, 28.4% were beneficial, 39.7% were neutral, and 31.9% were deleterious. It is important to mention here that, since the RNA polymerase is an essential gene, we cannot score null mutations, that usually account for less than 30% of all mutations. Therefore, the large number of beneficial mutations suggested a contrast between the protein-level and cellular-level impact of the mutation.**”*

Finally, I found what I think is a minor error I did not notice in my original reading of the manuscript, in the Introduction, starting at line 79: “The fitness costs associated with rifampicin resistance-conferring clinical RNAP mutations have led to the emergence of extreme drug resistance and multidrug resistance strains” - This sentence doesn’t make sense to me. Did the fitness costs lead to drug resistance, or did the mutations themselves?

We revised the sentence to make it clearer:

*“The fitness costs **of** the rifampicin resistance-conferring clinical RNAP mutations **may promote** the emergence of extreme drug resistance and multidrug resistance strains”*